# Recreating the biological steps of viral infection on a cell-free bioelectronic platform to profile viral variants of concern

Zhongmou Chao [1,3], Ekaterina Selivanovitch [1,3], Konstantinos Kallitsis [2], Zixuan Lu [2], Ambika Pachaury[1], Róisín Owens [2] & Susan Daniel [1] ✉

Viral mutations frequently outpace technologies used to detect harmful variants. Given the continual emergence of SARS-CoV-2 variants, platforms that can identify the presence of a virus and its propensity for infection are needed. Our electronic biomembrane sensing platform recreates distinct SARS-CoV-2 host cell entry pathways and reports the progression of entry as electrical signals. We focus on two necessary entry processes mediated by the viral Spike protein: virus binding and membrane fusion, which can be distinguished electrically. We find that closely related variants of concern exhibit distinct fusion signatures that correlate with trends in cell-based infectivity assays, allowing us to report quantitative differences in their fusion characteristics and hence their infectivity potentials. We use SARS-CoV-2 as our prototype, but we anticipate that this platform can extend to other enveloped viruses and cell lines to quantifiably assess virus entry.

RNA viruses tend to have higher mutation rates than their DNA counterparts[1–3], hence developing vaccines and antivirals that remain protective against disease-causing viral variants remains challenging with the fast pace of emerging variants of concern (VOC). When outbreaks of emergent viruses occur, quickly establishing the mechanisms of viral entry and transmission is critical for the rapid development of vaccines and therapeutics to combat RNA viruses and assessing emerging VOC and determining their potential for human harm. However, doing this is no easy feat; the mechanisms of viral infection are complex, involving numerous, multi-step biological processes, which often vary across cell types and microenvironments, hence necessitating a protracted incubation period for comprehensive data acquisition by live cell-based assays[4–6]. The entry pathway chosen by SARS-CoV-2, for example, is highly dependent on the interactions at the host cell membrane-virion interface, as well as the local protease, pH, and ionic conditions[7]. Because viral entry represents the first contact viruses have with host cells, the proteins and mechanisms that comprise these events have been targeted therapeutically and diagnostically to block or detect viral infection.

The last few years have witnessed a surge of advancements aimed towards the rapid, sensitive, and accurate detection of viruses and their emerging variants. While Reverse Transcription Polymerase Chain Reaction (RT-PCR)[8] remains the gold standard for detection, other classical methods include antibody detection[9], which relies on detecting antigen-specific antibodies in serum, and antigen detection[10], which uses designer antibodies to bind to and detect antigens. While these methods have been instrumental throughout the COVID-19 pandemic, they provide a binary response indicating either a detectable presence or absence of an antigen but offer few insights into their infectivity potential and are inadequate for screening VOC. Furthermore, studies have shown that as variants emerge, the ability of designed primers and antibodies to maintain their sensitivity diminishes, requiring the detection materials to be reformulated[11]. CRISPR-Cas- and isothermal amplification-based detection technologies have also been developed[12,13]. Techniques that fall into both categories detect nucleic acid (RNA or DNA) sequences and, while offering high sensitivity and selectivity, do not offer insights into a virus' structural integrity or its functionality. Biosensors, on the other hand, have been

[1]Robert Frederick Smith School of Chemical and Biomolecular Engineering, Cornell University, 124 Olin Hall, Ithaca, NY 14853, USA. [2]Department of Chemical Engineering and Biotechnology, University of Cambridge, Philippa Fawcett Dr., Cambridge CB3 0AS, UK. [3]These authors contributed equally: Zhongmou Chao, Ekaterina Selivanovitch. ✉e-mail: sd386@cornell.edu

shown to differentiate between a virus protein and a whole virus particle. They have been successfully used as detection platforms for coronaviruses by exploiting the specificity of antigens for their respective receptors[14–17]. However, to comprehensively understand the unique properties of emerging mutants and their potential for infection beyond mere binding interactions, a functional assessment of infectivity potential is imperative.

For enveloped viruses, which contain a lipid membrane that wraps or "envelopes" the genome-filled capsid, infection of the host cell is initiated when virions first bind to a host cell receptor, followed by the triggered fusion of the viral membrane with that of the host. These critical entry steps (binding and fusion) allow for the viral genome to be delivered to the host's cytosol. Chemically-responsive glycoproteins that protrude from the viral envelope mediate these entry processes. Their interactions with the cell plasma membrane receptors and other chemical cues create a fusion-promoting microenvironment. The cues that lead to viral entry typically involve some sequence of exposure to receptors, specific proteases, low pH, and ions. Depending on the host cell type, the identity of the triggers and the sequence of their presentation can vary. Additionally, the glycoprotein's properties (i.e., mutations that alter the glycoprotein in some way) also influence how they respond to these cues. Thus, it is a complicated convolution of glycoprotein sequence and host cell environment that controls the entry of these viruses and ultimately creates conditions for a productive infection of the host. Coronaviruses (including SARS-CoV, MERS and SARS-CoV-2), are a family of enveloped viruses and the variety of conditions that influence their biological entry pathways represent a major hurdle in probing viral infection mechanisms in vivo, as many methods lack the necessary control of the microenvironmental conditions and clear signals of a successful entry process. To gain the upper hand in mitigating future virus outbreaks and staying ahead of emerging VOC, it is crucial to develop platforms that are capable of both mimicking infection conditions and reporting infection progress via quantifiable readouts.

Here, we demonstrate the power of a technique that can detect viral entry processes but, importantly, provide quantitative readouts that distinguish entry characteristics of closely related viral strains. Starting with SARS-CoV-2 Wuhan-Hu-1 (WH1) as a model target, we present the design of an in vitro, cell-free entry platform (with a virus-free option as well) based on supported lipid bilayers (SLBs) that faithfully replicates the conditions that promote entry but in a convenient, controllable, and tailorable format with a much faster response time (~20 min) than live cell assays which usually take days. This cell-free platform senses entry functions electrically and is thus label-free. Next, we probe the entry characteristics of two SARS-CoV-2 Omicron subvariants, Omicron BA.1 and Omicron BA.4, and show that our platform identifies the known differences in fusion activity between these strains as well as repeats the known trends in their cell infectivity. This demonstration of using bioelectronics for detecting and characterizing virus-host entry processes is a critical precursor of the events that lead to host infection. Our device, mimicking the earliest events using "infection-on-a-chip", opens possibilities for examining entry conditions that can be leveraged for both basic science studies, screening assays for antiviral therapies, and fast assessment of entry characteristics that can inform the next steps in combatting VOC as they emerge.

## Results

### A description of the biological pathways of SARS-CoV-2 entry that are recreated in this platform

The exterior glycoprotein of SARS-CoV-2 is called Spike[18]. After the initial binding event between Spike and the host cell receptor (membrane-bound angiotensin-converting enzyme 2 (ACE2), viral entry continues via one of two entry pathways depending on its spatiotemporal exposure to microenvironmental cues[19]. Figure 1 briefly summarizes the two identified pathways for SARS-CoV-2 infection and the critical extra- and intracellular conditions that distinguish them, specifically focusing on the initial steps of binding to and fusion with the host cell's membrane. Which one of two entry pathways is triggered is cell type-specific and based on the availability of proteases for Spike cleavage. The first pathway, referred to here as the early entry pathway, is initiated when the transmembrane serine protease 2 (TMPRSS2) is present in the plasma membrane of the host cell[20]. Upon the binding of Spike protein to ACE2, the Spike is cleaved by TMPRSS2 to initiate virus-host membrane fusion, presumably at or near the cell surface, and the viral genome is delivered to the cytosol. The second pathway, referred to here as the late entry pathway, proceeds when the membrane-bound protease is not present in the plasma membrane of the host cell[21]. In this scenario, Spike protein binds to ACE2 and the virion is endocytosed, where it is subsequently cleaved by the endosomal cysteine protease-cathepsin L (CatL) inside the low pH microenvironment of the endosome. These cues trigger the fusion of the viral envelope with the endosomal membrane and release the genome into the cytosol.

### Design parameters for an Infection-on-Chip device

Taking inspiration from biological mechanisms, we set out to design a platform that can faithfully reproduce the microenvironments needed to selectively trigger either of the two entry pathways, and thus initiate the primary steps in a viral infection. There are four essential components in the construction of this infection-on-chip platform: (1) the presentation of viral and host cell membrane components, (2) spatiotemporal control over environmental cues required for triggering fusion, (3) a biocompatible scaffold accommodating membrane components for successful infection, and (4) quantifiable readouts reflective of different infection stages.

To test the infection-on-chip platform for its ability to recapitulate specific cell membrane environments that induce CoV entry events, we used Spike$_{WH1}$-incorporated viral pseudoparticles (VPP$_{WH1}$), produced using previously established methods[22]. VPPs are chimeric noninfectious particles and, in this case, derived from the MLV retrovirus. VPPs have been shown as successful and appropriate model systems for infectious viruses, including proteins found in SARS-CoV, SARS-CoV-2, and MERS-CoV (along with other examples from all three classes of fusion proteins)[23–26]. The exact components incorporated into the VPPs can be found in the Methods section. Confirmation of Spike protein incorporation is included in Supplementary Fig. 1. To capture the host cell features required for entry, but in a cell-free format, we selected SLBs to serve as host cell membrane mimics. These SLBs were composed of native cell membrane components (collected as plasma membrane blebs, or cell blebs) and "fusogenic" lipid vesicles, which self-assemble into a planar, single-bilayer lipid membrane blended with native cell membrane components, i.e., ACE2 receptors and TMPRSS2 proteases. The fusogenic vesicles used in this work are reconstituted from purified 1-oleoyl-2-palmitoyl-*sn*-glycero-3-phosphocholine (POPC) lipids, which are often used in the construction of biomimetic membranes. As shown in Fig. 1, this versatile, easy-to-assemble biomimetic membrane allowed spatiotemporal control over environmental cues to recapitulate both early and late entry pathways: when cell blebs containing ACE2 and TMPRSS2 (confirmed as shown in Supplementary Fig. 1) are incorporated into the SLB, colocalizing receptors and membrane proteases, the early entry pathway can be accessed. When SLBs are formed with only ACE2-containing blebs (confirmed as shown in Supplementary Fig. 1), only fusion via the late entry pathway can be activated when CatL is added under acidic conditions.

SLBs can be readily self-assembled on various functional supports, including biocompatible conductive polymers, which prompted our design of a label-free *SLB-on-electrode* structure to directly translate

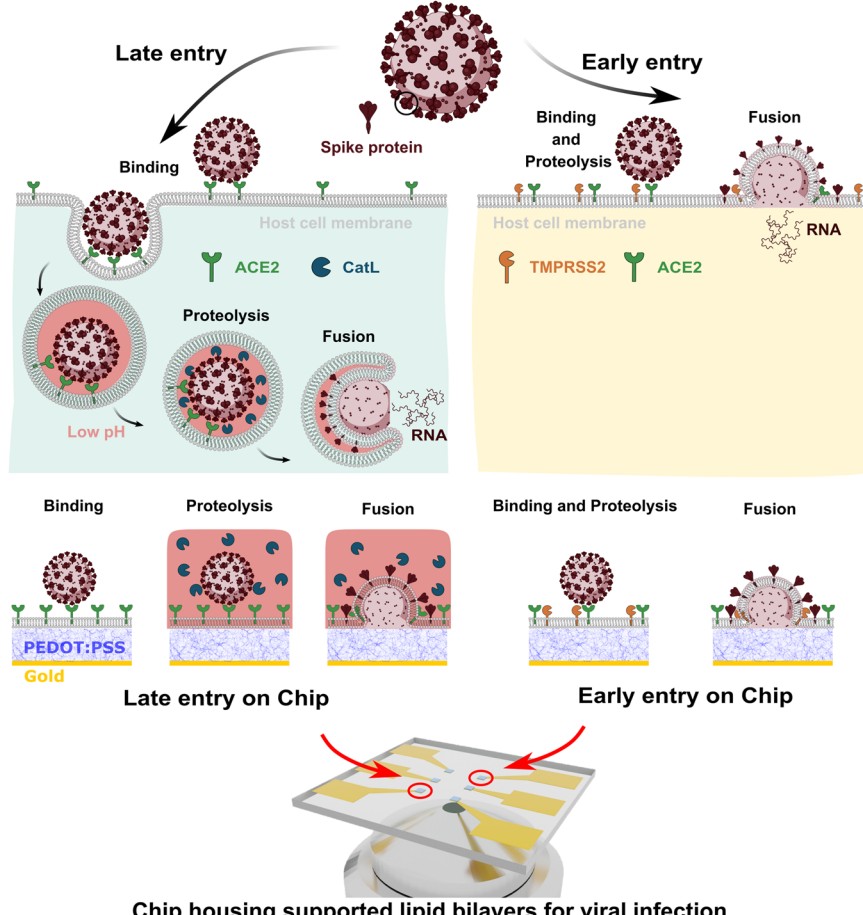

**Fig. 1 | SARS-CoV-2 entry pathways and the components needed to recapitulate these entry routes in an in vitro platform.** The two known pathways of SARS-CoV-2, including early entry, in which fusion is triggered by the TMPRSS2 protease, and late entry, in which virus particle fusion is catalyzed by the protease-cathepsin L (CatL) at low pH (note: pink color = acidic environment). We propose SLBs self-assembled on PEDOT:PSS electrode provide an ideal infection-on-chip platform. The SLB is formed using cell-derived blebs and fusogenic vesicles on a PEDOT:PSS surface, hence the membrane components are preserved. Viral pseudoparticles (VPP) with Spike protein, pH swap and soluble catalyst (CatL) can be included to induce fusion via the late pathway condition; including TMPRSS2 in the SLB triggers the early pathway condition. The optically transparent and conductive nature of PEDOT:PSS also allows both optical and electrical readouts to identify trends characteristic of binding and fusion events.

the interactions occurring at the biomimetic membrane into an electrical readout. Our group has previously demonstrated that SLBs can form on PEDOT:PSS (poly(3,4-ethylenedioxythiophene) polystyrene sulfonate) supports[27,28], a conductive, transparent polymer mixture widely used in biosensing applications[29,30]. We have also demonstrated that SLBs prepared on these polymer support maintaining two-dimensional fluidity of the constituents (both lipids and membrane proteins) and that the buffer-swollen polymer serves as a cushion that supports this characteristic of cell membranes in the resultant SLB[28].

Presented in the following sections, we fulfill all design parameters necessary for capturing SARS-CoV-2 infection-on-chip in a cell-free and label-free manner by building a biomembrane bioelectronic platform. We demonstrate that the label-free electrical readouts of this platform can provide a quantitative approach that could be used for investigating emerging variants, identifying potential VOC, and testing strategies to interrupt virus entry to thwart the progression of outbreaks. For example, the platform could be used as a tool to discover means to disrupt or arrest viral entry processes in antiviral drug development or in efforts to classify and differentiate properties of emerging variants as strains evolve, which can assist in predicting host tropism susceptibilities, or inform next-generation formulations of vaccines.

## Characterization of the Infection-on-Chip device

The sizes of $VPP_{WH1}$, cell blebs, and synthetic POPC vesicles were measured using dynamic light scattering (DLS) and nano particle analysis (NTA), and are reported to be ~100–200 nm, 150–450 nm, and 100 nm, respectively (Supplementary Fig. 2 and Supplementary Fig. 3). The particle counts, provided by NTA analysis, allowed us to control the relative concentrations of the POPC vesicles and blebs used for binding/fusion analysis and for SLB assembly respectively.

The method for forming SLBs from cell blebs and POPC vesicles on a PEDOT:PSS support is described in Methods. To assess their formation, we used fluorescence recovery after photobleaching (FRAP) measurements to confirm the formation of a mobile bilayer on PEDOT:PSS-coated glass coverslips–a critical prerequisite for the fusion events described later in this paper. For this optical characterization, SLBs formed on PEDOT:PSS surface were labeled with a lipophilic dye, R18, and in the case of a mobile bilayer, the R18 dye should diffuse freely throughout the SLB plane. Figure 2a shows typical FRAP data showing the full recovery of a photobleached spot on an SLB assembled using Vero E6 cell blebs and POPC vesicles. Vero E6 cells were chosen due to their endogenous expression of ACE2; therefore, the SLBs formed using blebs derived from this cell line incorporated the ACE2 receptor[31,32]. The other SLBs assembled for our study were derived from recombinant Vero E6 cells containing the TMPRSS2

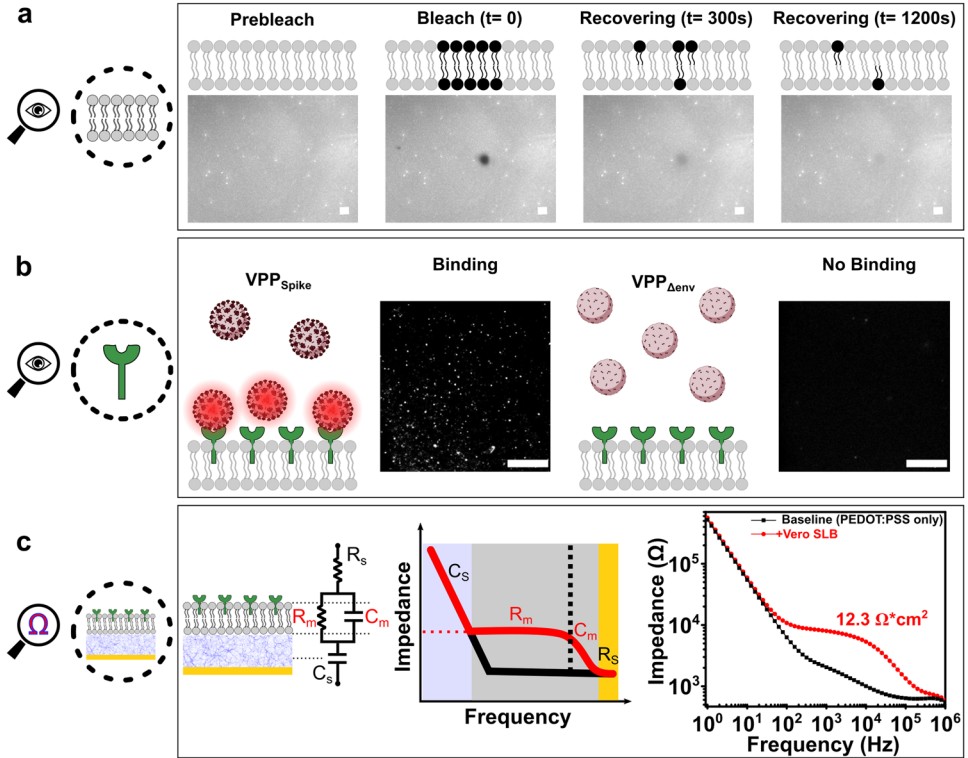

**Fig. 2 | Optical and electrical characterization of the infection-on-chip platform's components and functionalities. a** We used FRAP to characterize the mobility of SLBs formed on PEDOT:PSS surfaces. Shown here is one exemplary photobleached spot recovering over time from three independent experiments, indicating a mobile SLB. The cartoon representation is meant to provide a conceptual illustration of the technique. Indeed, our SLB was composed of both fluorescent and non-fluorescent lipids, and the fluorescence seen in the images is reflective of only the doped in R18 dye. **b** TIRF was used to confirm the existence of ACE2 receptor in SLBs: only fluorescently labeled VPP_Spike are visible at the SLB interface when bound to ACE2 receptors, while no fluorescently labeled VPP_Δenv were observed near the SLB due to the lack of binding interaction with ACE2 receptors on SLB. Shown here is one exemplary comparison from three independent observations. **c** EIS was used to characterize the electrical properties of an SLB on a PEDOT:PSS electrode. An SLB is modeled electrically as a capacitor and a resistor connected in parallel, hence its resistance ($R_m$) can be extracted by fitting to the RC(RC) circuit as shown. It can then be normalized by the area of electrode. All scale bars in this figure represent 20 μm.

receptor, and HEK293 cells used to assemble Spike-incorporating SLBs. We chose a TMPRSS2-modified Vero E6 cell line for the early entry pathway to remain consistent across as many parameters as possible for comparison with the late entry pathway using Vero E6 cells[33,34]. The FRAP images for these SLBs on PEDOT:PSS can be found in Supplementary Figs. 4 and 5. Upon photobleaching, the fluorescent intensity as a function of time in the photobleached spot was collected and fit with a Bessel function (see Methods) to later calculate the diffusion coefficient, D. All SLBs show comparable diffusion coefficients: ranging from 0.16–0.2 μm²/s and 0.92–0.99 mobile fractions (Supplementary Fig. 6).

To confirm the ACE2 receptors were incorporated into the SLBs, additional characterization of our SLBs was conducted using total internal reflection fluorescence (TIRF) microscopy. TIRF is an optical imaging technique especially suited to study the interactions occurring near the SLB-bulk interface, as its induced evanescent wave illuminates a limited (~100 nm) vertical region from this interface, effectively eliminating fluorescence from the unbound virus particles. Although our ultimate goal here is to validate a label-free sensing platform for viruses, the visualization of binding events between ACE2 and VPP_WHI was necessary to verify that native cell receptors from blebs were incorporated into the SLB assembled on PEDOT:PSS. VPP_WHI were labeled with R18 fluorophores that partition into the VPP membrane envelope; the SLB was not labeled in this experiment. We used TIRF microscopy to measure the interaction between ACE2 receptors in the SLBs and fluorescently labeled VPP_WHI. As shown in Fig. 2b and Supplementary Fig. 7, a representative TIRF field of view

(FOV) suggests that the R18-labeled VPP_WHI are bound to the ACE2 assembled SLB, while particles devoid of Spike proteins (VPP_Δenv) do not exhibit any detectable signals, as a control case. A similar observation can be made when VPP_WHI are introduced to SLBs without ACE2 proteins (see Supplementary Fig. 8). These data together suggest that no binding interactions are observed in the absence of either Spike protein or ACE2 receptor.

PEDOT:PSS is not only conductive, it is also a volumetric capacitor—making it an ideal electrode material for electrochemical impedance spectroscopy (EIS) measurements as it significantly reduces system impedance[35]. Lower system impedance enables the measurement of small changes in SLB electrical properties that can be correlated with viral entry processes, as we describe later. EIS is a non-invasive electrical sensing technique with a proven track record for accurately quantifying bio-recognition events occurring at biointerfaces[36–38]. When an SLB is self-assembled on PEDOT:PSS electrodes, the ionic flux reaching the electrode surface is reduced due to SLB shielding, thereby decreasing the ionic current. This outcome is ultimately measured by an increase in circuit impedance when compared to the electrode baseline signal (a circuit without SLB coating). The PEDOT:PSS electrodes used in this work were defined on a gold contact pad using optical lithography (see Methods). As shown in Fig. 2c, when no SLB was formed on the PEDOT:PSS electrode, the frequency-dependent impedance (plotted in black), generates a "hockey stick" shape typical of a bare electrode baseline signal (PEDOT:PSS only) of a resistor-capacitor in series structure. Upon the self-assembly of a SLB on the electrode surface (+Vero

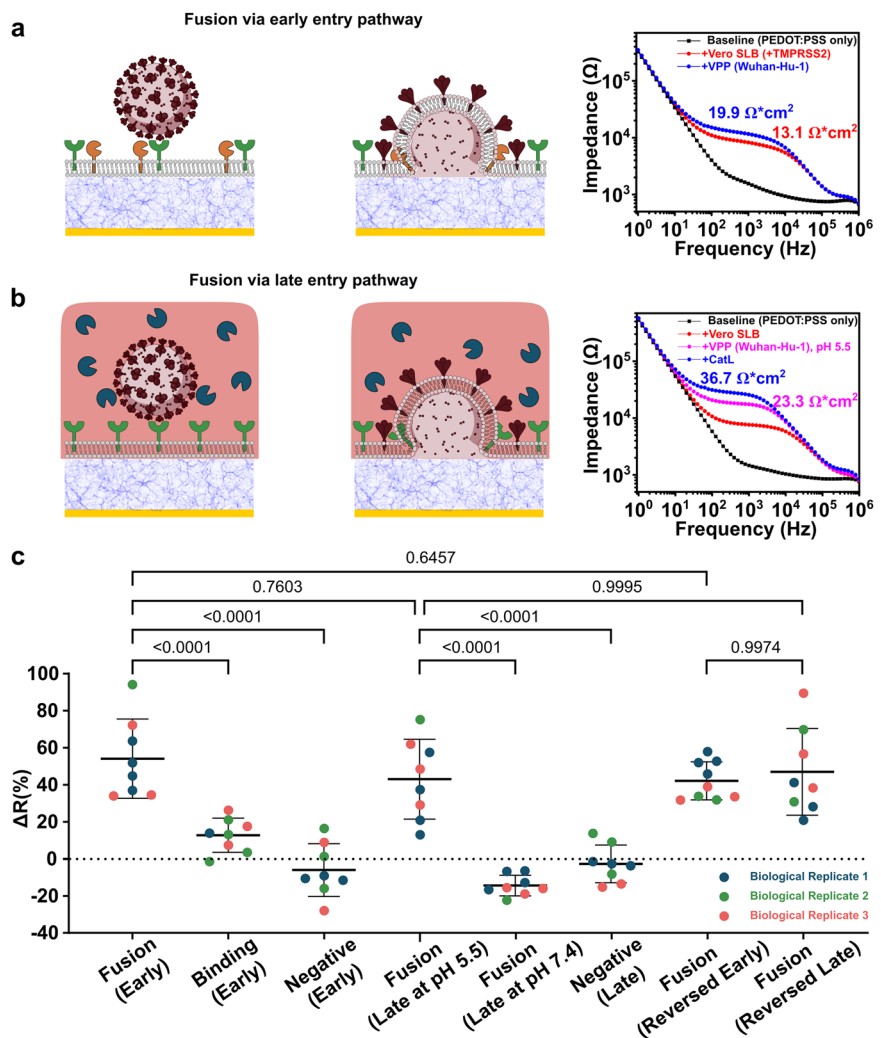

**Fig. 3 | Electrical responses of fusion via early and late entry pathways recapitulated on host cell-derived SLB. a** The experimental group for early fusion pathway consisted of VPP$_{Spike}$ and an SLB containing ACE2 (green) and TMPRSS2 (yellow). EIS Signals are characteristic of fusion events showing the changes in SLB membrane resistance in the equivalent electrical circuit scenario; **b** the experimental group consisted of the VPP$_{Spike}$ and an SLB containing ACE2 (green) and CatL (navy), where signals are characteristic impedance data of fusion events (note: pink color = acidic environment); **c** distribution of membrane resistance changes at all events of all systems (3 biological replicates for all systems, $n = 9$ for reversed early fusion and $n = 8$ for all other systems) using Spike protein from SARS-CoV-2 Wuhan-Hu-1. ΔR data are mean ± SD; statistical analysis was performed using one-sided one-way analysis of variance (ANOVA) with Šidák's multiple comparisons test.

SLB), the circuital response shifted from black "hockey stick" data trend to the red "chair shape" data trend, confirming the addition of a resistor-capacitor in parallel to the overall electrical circuit. RC is an established electrical trait of lipid bilayers[39,40]. The membrane resistance ($R_m$) and capacitance ($C_m$) of the SLB can then be extracted by fitting the signal to an equivalent electrical circuit, as depicted, and then normalized by the area of the electrode (see Methods).

### Recreating the SARS-CoV-2 entry pathways using Infection-on-Chip

Now that we have formed a mobile SLB with confirmed native membrane components using FRAP, demonstrated that the initial step in SARS-CoV-2 infection process (i.e., binding) using TIRF, and verified that SLB formation on the PEDOT:PSS electrode results in a measurable signal using EIS, we continued to investigate if fusion can be initiated and detected on our chip when environmental cues are integrated spatiotemporally.

We first focused on the early entry pathway, where we mimicked the respective fusion triggering environment by forming an SLB from cell blebs containing ACE2 and TMPRSS2 on the PEDOT:PSS surface. We then introduced the VPP$_{WH1}$ to monitor binding and fusion as

depicted in the schematic shown in Fig. 3a. Electrical readouts were conducted on PEDOT:PSS electrodes. As expected, when SLBs were formed on the electrodes, the electrical circuital response to alternating voltage shifted from the black (PEDOT:PSS only) to the red (SLB) signal, as shown in Fig. 3a (right). Subsequently, upon the addition of the VPP$_{WH1}$ to the SLB with both ACE2 receptors and TMPRSS2 proteases, the circuital response shifted from red to blue and, when fit and normalized, the SLB membrane resistance increased from 13.1 to 19.9 $\Omega \times cm^2$ (+51.9%), while no substantial membrane capacitance change was observed (see Supplementary Table 1. This table lists all SLB membrane capacitances measured, which range from 0.9–1.8 μF/cm², corresponding to a parallel plate capacitor filled with a dielectric and separated about the same distance as the thickness of a typical SLB, 4–5 nm. We hypothesize that the increase in resistance is attributed to the incorporation of additional biomacromolecules originally in the viral envelope now present in the SLB after the fusion event takes place−an observation that is consistent with optical data under the same conditions (Supplementary Fig. 7). VPP$_{WH1}$ were also added to a SLB containing ACE2 (no TMPRSS2 protease) to measure the electrical response arising only from binding interactions, while VPP$_{Δenv}$ were added to a SLB with both ACE2 and TMPRSS2 to identify any

non-specific interactions between the bilayer and pseudoparticles not directed via Spike-ACE2 binding. The electrical responses from both control groups are consistent with optical data, as shown in Supplementary Fig. 7, suggesting minimal interactions when compared to the conditions that promote fusion. The differences in both electrical and optical signals between binding and fusion events suggest that we can differentiate between them under conditions suitable for the early entry pathway using the electronic label-free approach on our infection-on-chip devices.

The late entry pathway requires protease CatL, instead of TMPRSS2, to catalyze the virus-membrane fusion. Biologically, coronavirus entry in the late pathway can proceed via several different routes including clathrin dependent, caveolae dependent/clathrin independent, or caveloa/clathrin independent[21,41,42]. Since our platform does not mimic the endocytosis mechanisms, we reproduce this pathway using our model system by supplementing the bulk solution with CatL, which is a soluble protein, and mimicking the acidic endosomal environment in which CatL is active. To mimic this environment and triggering conditions in our platform, we generated SLBs made from Vero E6-derived blebs, which contained the ACE2 receptor but no TMPRSS2. To recreate the endosomal triggering environment, as shown in Fig. 3b, we exchanged the initial pH 7.4 buffer for a more acidic buffer (pH 5.5) and then added soluble CatL, which is active at pH 5.5 but not at pH 7.4. Similar to the early entry pathway, the electrode baselines were acquired before SLB formation (black) and after SLB formation (red), as shown in Fig. 3b (right). $VPP_{WH1}$ were first added to bind with the ACE2 receptors in the SLB at pH 7.4, before exchanging the buffer to a more acidic environment (pH 5.5). As a result, the electrical signal shifted from red to pink, indicating that the binding between ACE2 receptors and the VPP, together with the pH drop, contributed to an increase in membrane resistance, aligned with our observation in the early entry pathway (Supplementary Fig. 7) and previous reports[37]. Upon the addition of CatL, the SLB membrane resistance further increased (blue trace, from 23.3 to 36.7 $\Omega \times cm^2$, + 57.5%), suggesting successful fusion between the VPPs and SLB membranes. As a control for fusion at non-optimal triggering conditions, CatL was also added to a non-acidic buffer environment after the $VPP_{WH1}$ addition. The electrical signal (Supplementary Fig. 9) suggested membrane resistance dropped insignificantly, indicating there was no fusion due to non-optimal triggering conditions; $VPP_{\Delta env}$ were used as a negative control, and no significant membrane resistance shift was observed at lower pH after the addition of CatL (Supplementary Fig. 9). These measurements are all congruent with the optical data (Supplementary Fig. 9). From the electrical and optical data, it is clear that both CatL and acidic conditions are required for promoting fusion of the $VPP_{WH1}$ with the SLB, an observation that is consistent with our current understanding of SARS-CoV-2 viral entry[43,44].

The repeatability over biological and technical replicates of electrical responses for fusion and control groups for both pathways is shown in Fig. 3c. The change in resistance values for fusion events of $VPP_{WH1}$ are comparable in the early (+54.0 ± 20.0%) and late entry (+42.9 ± 20.2%) pathways, both distinct from all control groups.

## Differentiating between Wuhan-Hu-1, Omicron BA.1, and BA.4 strains using *Infection-on-Chip*

The $VPP_{WH1}$ were used in all the entry experiments so far, and we have confirmed both entry pathways can be recapitulated using the Infection-on-Chip platform. Next, we investigated if our platform was capable of distinguishing SARS-CoV-2 variants with different fusogenicities. Omicron BA.1 and BA.4 (BA.1 and BA.4) were selected in this study since BA.1 has been reported to be less fusogenic than BA.4, while both Omicron variants selected have lower fusogenicities than the WH1[45–47]. In this case, fusogenic properties refer to the relative number of pseudoparticles fusing with the ACE2-containing membranes.

The electrical readouts modeling the early and late entries of BA.1 are shown in Fig. 4a. We see no significant membrane resistance increase in the case of early pathway (left), yet a small, but distinguishable resistance increase can be observed in the case of late pathway (middle). Statistical data (right) suggested significance between early and late entries of BA.1, matching recent reports[46,48–51].

The resistance values of both early (left) and late (right) entries of BA.4 are shown in Fig. 4b. Comparing BA.1 to BA.4 VPP, membrane resistance increases were more significant for $VPP_{BA.4}$, as suggested by statistical data (right), supporting the reports of BA.4 being more fusogenic than BA.1[45,52]. However, when comparing wild-type SARS-CoV-2 to the BA.4 strain, shown in Fig. 3c, the membrane resistance increase caused by the fusion of $VPP_{BA.4}$ was still significantly reduced: from +54.0 ± 20.0% to +21.4 ± 10.3% for the early entry pathway and from +42.9 ± 20.2% to +24.6 ± 12.1% for the late pathway. Our results matched strongly with viral transduction assays, as shown in Fig. 4c, where the relative luciferase units detected using the $VPP_{WH1}$ were about 7× higher than $VPP_{BA.1}$ and 4× higher than $VPP_{BA.4}$. A detailed description of the transduction assay is provided in the Methods section. Our EIS-based fusion assay aligns well with other standard assays used to determine the relative infectivity of virus particles, such as syncytia and plaque assays, evaluating the relative fusogenicities of the three variants explored here[45–47], confirming the accuracy of Infection-on-Chip platform in distinguishing SARS-CoV-2 variants and importantly in a timescale unmatched by traditional cell-based assays (minutes versus days).

## Reversing SARS-CoV-2 early and late pathway configurations

The previous arrangements used the SLB as a model for either the cellular or endosomal membrane surfaces and the VPP as mimics of the infectious virus. Here, we swap the active constituents of both entry pathways, where now the SLB displayed features found on the virus surface (i.e., the glycoproteins), while blebs in the bulk phase presented their respective host cell surfaces. Specifically, we constructed SLBs that contained $Spike_{WH1}$ protein and formed cell blebs that contained ACE2 or ACE2/TMPRSS2 as host cell "particles" that can bind to and fuse with the Spike-containing SLBs. By swapping the constituent presentation, we present a strategy for rapidly screening cell types and their respective susceptibility to viral infection without the need for virus particles (virus-free) and only the spike protein gene for cellular expression.

Spike proteins were incorporated into the SLB by rupturing Spike-transfected HEK293 blebs, while TMPRSS2-modified Vero E6 cell blebs (Fig. 5a) and Vero E6 cell blebs (Fig. 5b) were introduced to evaluate and quantify their interactions with the "virus-like" SLB. Mirroring our previous experiments, the SLBs were formed on both PEDOT:PSS-coated glass coverslips (Supplementary Fig. 10) and PEDOT:PSS electrodes for optical and electrical readout, respectively.

We first investigated the electrical responses for reversed "early entry". Similar to the more traditional display of constituents described earlier, the electrical signal shifted from the electrode baseline (black) to the SLB signal (red) (Fig. 5a). After the addition of cell blebs with ACE2 receptors and TMPRSS2, Spike SLB membrane resistance increased from 20.3 to 31.0 $\Omega \times cm^2$ (+52.7%), showing a similar membrane resistance increase as measured in early entry pathway as shown in Fig. 3a, c. Similarly, in the reversed "late entry", cell blebs with ACE2 were added to bind with the Spike SLB and soluble CatL was added to initiate the fusion, after swapping to an acidic buffer environment. The electrical response at each step was measured and plotted in Fig. 5b. After the formation of Spike SLB (red), membrane resistance increased to 51.7 $\Omega \times cm^2$ (pink) upon binding with ACE2-containing blebs and exchanging to a lower pH buffer environment (from PBS pH 7.4 to PBS pH 5.5). Membrane resistance increased to 72.1 $\Omega \times cm^2$(blue) after the addition of CatL (+39.5%), comparable to the electrical response of the late entry pathway shown in Fig. 3b, c. This work shows the SLB based

 

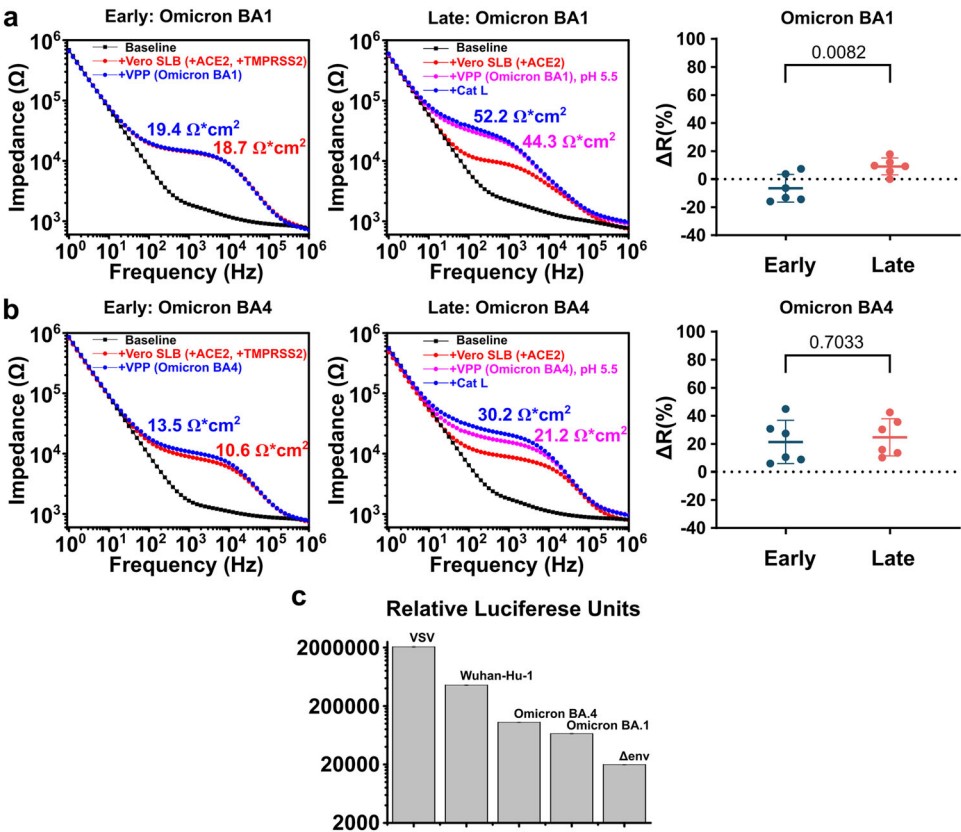

**Fig. 4 | A comparison of viral fusogenicities of SARS-CoV-2 VOC by electrical response and viral transduction assays. a** EIS electrical signal change of fusion via early (left), late (middle) entry pathways and their statistical comparison (right, $n = 6$ for both pathways) using Omicron BA.1 VPP; **b** EIS electrical signal change of fusion via early (left), late (middle) entry pathways and their statistical comparison (right, $n = 6$ for both pathways) using Omicron BA.4 VPP. ΔR data are mean ± SD; statistical analysis was performed using a two-sided unpaired $t$ test; **c** relative transduction efficiencies of the Wuhan-Hu-1 Spike and Omicron variant Spike-containing pseudoparticles. The transduction efficiency of $VPP_{Spike}$ was assessed

against a positive control that contained a vesicular stomatitis virus G protein ($VPP_{VSV}$) and a negative control without any envelope protein ($VPP_{\Delta env}$). The luciferase production of the infectious $VPP_{Spike}$ and $VPP_{VSV}$ was consistently orders of magnitude higher than $VPP_{\Delta env}$, indicating that the particles we produced were "active" and capable of fusion with a cell membrane. The samples labeled BA.1 and BA.4 refer to Omicron variants. All infectivity assays were completed with Vero E6 TMPRSS2 cell lines. All data above represent five technical replicates ($n = 5$). Error bars represent standard deviation.

infection-on-chip platform can be used to quickly screen interactions between Spike proteins and host cell membranes without producing VPP or virus-like particles (VLP). This can be especially useful for screening antibodies against Spike protein and small molecule fusion inhibitors in a high throughput manner. In the next section, we provide an example of how our platform can be used to study protease inhibitors in the same spirit.

## Suppressing SARS-CoV-2 fusion with SLB via the usage of protease inhibitor

One effective strategy to block SARS-CoV-2 infection is targeting viral entry into host cells[53,54]. Here, we investigate if the fusion of $VPP_{Spike}$ can be blocked using a protease inhibitor. Camostat mesylate is effective in blocking SARS-CoV-2 early entry into host cell by inhibiting TMPRSS2 protease[55]. To assess the impact of camostat mesylate blockage of TMRRSS2 in SARS-CoV-2 early entry, we formed SLB from native blebs containing both ACE2 and TMPRSS2. However, because our assay is cell-free, we tested two ways to introduce this drug into the assay. In the first approach, after the initial SLB formation on PEDOT:PSS electrode, camostat mesylate (50 µM) was incubated with the SLB for 2 h; in the second approach, we incubated freshly harvested vero blebs (+ACE2, +TMPRSS2) with 50 µM of camostat mesylate overnight *prior* to SLB formation on PEDOT:PSS electrode. In both approaches, $VPP_{WH1}$ were then added to trigger fusion and SLB resistance monitored to determine if fusion was able to proceed. As a

control and for comparison, the SLB resistance change caused by $VPP_{WH1}$ fusion in the absence of camostat mesylate was also collected.

To assess the fusion blockage of camostat mesylate, the resistance change of two experimental groups were normalized with respect to resistance change in the control group and is plotted in Fig. 6. In the first case, camostat mesylate incubation with SLB, the change in SLB resistance due to $VPP_{Spike}$ fusion is significantly reduced, showing an 87.3% reduction with respect to no camostat mesylate positive control as shown in Fig. 6. We conclude this is due to the suppression of TMPRSS2 function in mediating viral entry through early pathway spike cleavage. Similarly, in the second case, when incubating camostat mesylate with cell blebs prior to SLB formation, the SLB impedance change was only 30% relative to the positive control case without camostat mesylate, indicating a 70% fusion reduction. These electrical results support that the TMPRSS2 inhibitor camostat mesylate significantly blocks SARS-CoV-2 fusion in our platform, consistent with previous reports[19,55], demonstrating a possibility for using this cell-free platform to screen therapeutics targeting SARS-CoV-2 entry.

## Discussion
### Infection-on-chip model
The Infection-on-Chip devices operate by detecting changes in bio-membrane electrical properties to characterize specific interactions between the VPP and the host cell membrane that occur when the viral infection process begins. To achieve this, the devices were constructed

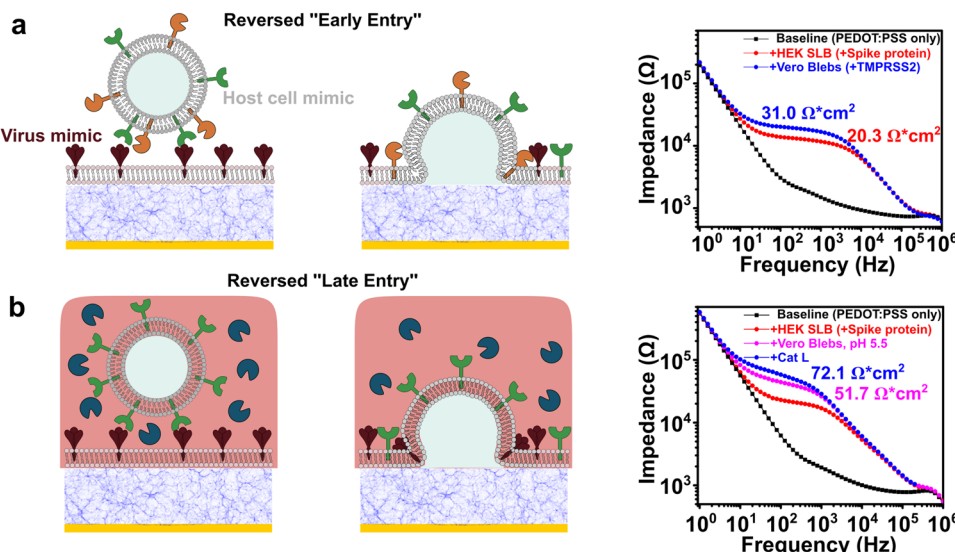

**Fig. 5 | Reverse geometry fusion studies using SLB containing Spike protein with different blebs. a** Illustration of "reversed" early entry pathway and its corresponding EIS electrical signal change; **b** "reversed" late pathway and the corresponding EIS electrical signal change.

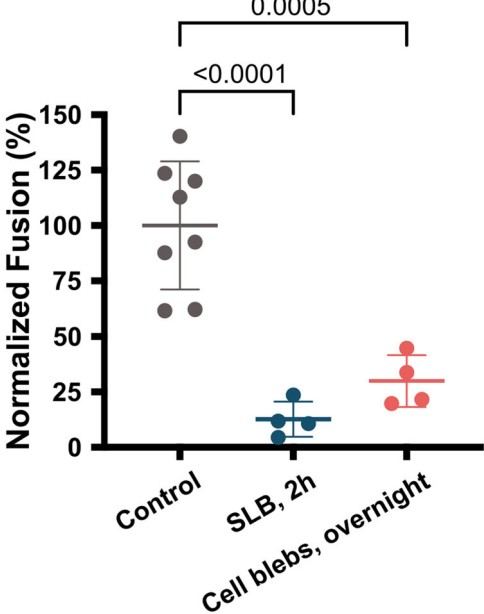

**Fig. 6 | A comparison of SLB (+TMPRSS2, +Vero) and VPP_WHI fusion results under different blocking conditions with TMPRSS2 inhibitor (camostat mesylate).** Fusion results of camostat mesylate incubating with SLB for 2 h ($n = 4$) and camostat mesylate incubating with cell blebs overnight ($n = 4$) are normalized as SLB membrane resistance change after fusion with respect to the control group ($n = 8$), in which no camostat mesylate was used; statistical analysis was performed using one-sided ordinary one-way analysis of variance (ANOVA) with Tukey's multiple comparisons test.

from the necessary biological and chemical elements described earlier and responses were modeled using electrical components of resistors and capacitors. The most rudimentary system resulted in signal contributions from the electrolyte solution resistance and PEDOT:PSS capacitance (electrode baselines in Figs. 2–5) and resistance and capacitance elements derived from the presence of a membrane coating the PEDOT:PSS electrode. The resistance fluctuations of the RC circuit elements from the SLB are used as a diagnostic tool to distinguish between only binding and binding plus fusion events, the data for which are presented in Figs. 3–5.

Although we have not yet identified the mechanism by which binding and fusion lead to an increase in resistance, we hypothesize that it could be via two potential mechanisms: (1) as more material is integrated into the SLB, the increase in protein and lipid density results in an increase in resistance due to tighter packing and a change in material properties, or (2) as more proteinaceous and lipid materials are added to the SLB, membrane defects, or "gaps", are filled in, consequently increasing the resistance of the film itself. Both hypotheses are evinced in the fusion pathways for fusogenic Wuhan-Hu-1, where resistance values increased by 40–60% from the baseline. Conversely, the less significant resistance increases upon binding, shown in Fig. 3c and Supplementary Fig. 7, support the second hypothesis. In this scenario, the ACE2 and VPP interactions result in VPP immobilization proximal to the SLB surface, potentially blocking defects near the binding site, without integrating into the SLB itself. Though we do not know the exact mechanism that leads to the change in resistance, overall, we can reproducibly identify and characterize fusion events, which can be especially beneficial for isolating, particularly infectious viral variants or screening for therapeutics that target either event.

### SARS-CoV model system

The two known entry pathways of SARS-CoV-2 capture the canonical features of coronavirus' initial infection stages, making it an excellent model system for our study. Though the specific receptors and required triggers vary between viral strains and species, there are fundamental aspects that are conserved. For example, there are currently seven identified human coronaviruses (hCoV), among which the most notable are SARS-CoV and MERS-CoV. Though SARS-CoV and SARS-CoV-2 entry mechanisms share more similarities, both requiring an ACE2 protein for binding, all three coronaviruses (SARS-CoV-2, SARS-CoV, and MERS-CoV) share similar fusion mechanisms via TMPRSS2 or CatL activation. Going beyond the Coronaviridae family, viruses from the Orthomyxoviridae and Rhabdoviridae families, such as influenza and VSV respectively, also share similarities with the late entry pathways of SARS-CoV-2, requiring an acidification step to prompt fusion. The Infection-on-Chip platform may be leveraged to easily identify cell types particularly susceptible to each virus, provide mechanistic information into the events that initiate infection, and evaluate differences between emerging variants.

The SARS-CoV-2 model system provided an opportunity to determine whether or not the platform can detect variability between different Spike protein variants. Since Omicron variants are now dominant globally, we produced Omicron Spike-incorporating VPP, compared to the Spike$_{Wuhan-Hu-1}$ proteins used for the initial experiments. Experiments were first conducted on Omicron BA.1 variants, which showed a decrease in fusion activity, as evidenced by a decreased change in resistance for both early- and late-entry pathways (Fig. 4a), albeit a less significant decrease was observed for the late-entry pathway. This was consistent with recently published findings in which it was identified that Omicron BA.1 and BA.2 variants exhibit an altered entry preference compared to ancestral SARS-CoV-2: preferring endosomal (late) entry pathway as these Omicron variants are less dependent on the TMPRSS2 protease[46,48–51]. Since there are several Omicron variants that have emerged, each with unique sets of mutations, we also evaluated a more fusogenic variant—Omicron BA.4. Our data correlated well with these reports, as the change in resistance increased for both pathways when using VPP incorporating Spike$_{Omicron BA.4}$ (Fig. 4b). Our data was not only analogous to existing reports of entry-pathway preference, we were also able to detect fusion variability between WH1, Omicron BA.1, and Omicron BA.4 variants that directly mirrored those reported[45]. In these reports the WH1 exhibited the highest fusogenicity, followed by Omicron BA.4, and Omicron BA.1 as the least fusogenic of these mutants. These distinctions further highlight the benefits of using this platform with electrical sensing for straightforward screening of viral mutants, and use the acquired data to distinguish highly infectious mutants from those that are less infectious.

Lastly, the Infection-on-Chip platform is capable of distinguishing between binding and fusion events using EIS by introducing different environmental cues that support either binding or fusion. For example, for the late entry pathway the initial increase in resistance is indicative of binding only, as no fusion trigger was present and, by rinsing the reaction well, we negated any transient interactions between the SLB and the VPPs. This interpretation was then verified using TIRF where fluorescently labeled VPPs can be seen bound to the ACE2-containing SLBs, but not in the case of the negative controls (Supplementary Fig. 8). As the CatL protease was added to the reaction, fusion was triggered and detected using another increase in resistance. If there was no binding, the initial increase would not have been detected and, if there was binding but no fusion, the subsequent increase would not have been detected. The environment for the early entry pathway can likewise be controlled except in this case, the protease fusion trigger is located within the SLB. To decouple the binding from fusion in this case, two possibilities are available to the experimentalist, which are commonly used in cell-based assays: low temperature binding to limit fusion, then an increase in temperature to induce it. Alternatively, one could use various drugs to temporarily inhibit protease cleavage until binding is complete, then remove the drug to activate the protease. This platform allows the experimenter to envision and execute a series of conditions and environmental cues customized to determine binding *versus* fusion, study one pathway over the other, or vary any other aspect of the system that would be difficult or nearly impossible in the cell-based system, not to mention doing so in the fraction of time needed for standard biological transduction assays.

## Prospects

"Bioprocesses"-on-chip devices, such as cell-on-a-chip, organ-on-a-chip, and tissue-on-a-chip, for instance, represent emergent platforms of interest amongst the biomedical and biomaterials community. Among a myriad of other benefits, their recent successes as in vitro micro-scale physiological models can potentially transform fields that focus on therapeutic development and personalized medicine. Our proposed infection-on-chip platform complements these existing technologies by providing mechanistic information at the membrane level without relying on downstream effects or signals. In other words, our readouts directly correlate to events at the membrane-virus interface with exquisite control over the participating components (i.e., receptors, environmental conditions, and presented pathogens), how they are presented, and the functionality of the participating constituents (i.e. either binding or fusion events between the SLB and VPP). Whether using the more traditional display, in which the SLB mimics the cellular surface or a presentation where the SLB emulates the viral surface, the infection-on-chip platform can be employed as a quantitative scaffold to interrogate biological pathways or as a tool to rapidly screen interactions with a viral or cellular surface, both of which should assist determining societal responses as VOC continue to emerge.

## Methods

### Materials

The 1-palmitoyl-2-oleoyl-sn-glycero-3-phosphocholine (POPC), used for the preparation of fusogenic liposomes, was purchased from Avanti Polar Lipids (700 Industrial Park Dr, Alabaster, AL 35007). Biotechnology-grade chloroform was used during the preparation of the POPC liposomes and was purchased from VWR (1050 Satellite Blvd. Suwanee, GA 30024). Whatman Nucleopore polycarbonate filters (50 nm) (Cytiva- Marlborough, MA) were used for liposome extrusion. The octadecyl rhodamine B chloride (R18), used as a lipophilic dye for collecting optical data, was made by Invitrogen purchased from Thermo Fisher Scientific- Waltham, MA. Dulbecco's Modified Eagle Medium (DMEM) was used as a basal medium for cell growth and to produce pseudoparticles, along with Gibco Fetal Bovine Serum (FBS) and Gibco Penicillin-Streptomycin (10,000 U/mL) when indicated. TurboFect, Lipofectamine 2000, and Gibco Opti-MEM were purchased through Life Technologies Thermo Fisher and were used for the necessary transfection protocols described later in this section. Corning Trypsin 1×, 0.25% Trypsin purchased through VWR, 0.53 mM EDTA in HBSS [-] calcium, magnesium was used as the enzymatic agent during passaging. 4-(2-Hydroxyethyl)piperazine-1-ethanesulfonic acid (HEPES), dithiothreitol (DTT), and formaldehyde solution, used for the preparation of the blebs, were all purchased from MilliporeSigma. VWR 25 mm × 25 mm glass coverslips were used for the preparation of the supported lipid bilayers and as solid supports for the collection of optical data. The Piranha wash consisted of sulfuric acid (95- 98%, VWR) and hydrogen peroxide (50 wt.% solution, Krackler Scientific). PEDOT:PSS (PH 1000) was purchased from Ossila (Sheffield, UK), (3-Glycidyloxypropyl)trimethoxysilane (GOPS) was purchased from MilliporeSigma. The following antibodies were obtained: ACE2 Antibody, Supplier- Cell Signaling Technology, Polyclonal Antibody, Species-Rabbit, Concentration- 220 µg/mL, Dilution- 1:500; TMPRSS2 Antibody, Supplier-NovusBio, Polyclonal Antibody, Species- Rabbit, Dilution- 1:500; SARS-CoV-2 Spike Antibody, Supplier- Sino Biological, Polyclonal Antibody, Species- Rabbit, Dilution- 1:1000. ImageJ 1.53 A, AxioVision rel. 4.8, Zen 3.4 were used to acquire and analyze optical data. NOVA 2.1.7 was used to collect electrical data. Igor Pro 9, Origin 2016, and Prism 10 were used to plot data found in the main and supporting figures.

Buffers and other solutions:

GPMV Buffer A: 2 mM $CaCl_2$, 10 mM HEPES, 150 mM NaCl, pH 7.4

GPMV Buffer B: 2 mM $CaCl_2$, 10 mM HEPES, 150 mM NaCl, 25 mM formaldehyde, 2 mM DTT pH 7.4

Reaction Buffer A: 137 mM NaCl, 2.7 mM KCl, 10 mM $Na_2HPO_4$, 1.8 mM $KH_2PO_4$, pH 7.4

Reaction Buffer B: 137 mM NaCl, 2.7 mM KCl, 10 mM $Na_2HPO_4$, 1.8 mM $KH_2PO_4$, pH 5.8

C-DMEM: DMEM, 10% (v/v) FBS, Penicillin-Streptomycin (200 units/mL and 200 µg/mL)

F-DMEM: DMEM, 10% (v/v) FBS

## Cell culture

African green monkey kidney cells (Vero E6) from ATCC, TMPRSS2 enhanced Vero E6 from the JCRB Cell Bank, and Human embryonic kidney cells (HEK-293T) from ATCC were maintained in C-DMEM at 37 °C in an incubator containing 5% $CO_2$ and 95% air. All cells were passaged upon reaching 80–95% confluency by first washing the cells with Dulbecco's phosphate-buffered saline (DPBS) and then enzymatically releasing them from the flasks using Trypsin EDTA 1×. Confluency was monitored using bright-field microscopy. Sample sizes for cell culture were determined by surface availability for optimal confluency. Cells were counted in order to determine the optimal number to seed in accordance with standard biological known values.

## GPMV ('bleb') preparation

GMPVs were prepared using previously established methodologies aimed at producing free GMPVs from attached cells. Once the cells have achieved >90% confluency, in preparation for blebbing, the cells were washed with GMPV buffer A (3×). Freshly prepared GPMV Buffer B was then added to the plate and incubated at 37 °C for 2 h. Both GPMV Buffer A and GPMV Buffer B contain small amounts of $CaCl_2$, as calcium has been found to be crucial for promoting an optimal fusion environment[23,56]. The buffer, now containing the GPMVs, was decanted into a conical tube and incubated on ice for 45 min. Post incubation, the top 80% of the solution was collected, and the bottom 20% was disposed. The GMPVs were characterized using DLS using a Malvern Panalytical (Enigma Business Park, Grovewood Road Malvern, WR14 1XZ, UK) Zetasizer and NanoSight. The concentrations of blebs was always $10^9$ particles/mL and NanoSight measurements over a two-week period can be found in Supplementary Fig. 3. Fresh GMPVs were prepared every two weeks to ensure that maximum protein activity was maintained.

## Preparation of pseudotyped particles

Human embryonic kidney cells HEK293 cells were seeded on 6-well plates with 2 mLs of C-DMEM solution per well. The cell density typically reached ~50% confluence prior to proceeding to the next step. Transfection was performed with three plasmids encoding for the different proteins required to form pseudotyped particles: the envelope glycoprotein, MLV gag and pol proteins, and luciferase reporter. The total amount of DNA per well was 1 µg with 300 ng of Gag-Pol, 400 ng of luciferase reporter, and 300 ng of the envelope protein (all sequences encoding for the genes can be found in Supplementary Table 2). First, the plasmids encoding for Gag-Pol and luciferase were combined and incubated at room temperature for 5 min in an Eppendorf tube. For a 50 mL solution, 1.25 mLs of optimem and 1.4 mLs of polyethyleneimine (PEI) were added to a 50 mL falcon tube. The plasmids for the envelope proteins were added to the Falcon tube as well, appropriately scaling the amount to the 50 mL total volume. The envelope proteins were either SARS-CoV-2 spike protein, vesicular stomatitis virus (VSV) G glycoprotein, or a negative control that lacked any enveloped glycoproteins (Δenv). The backbone proteins (Gag-Pol) and luciferase plasmids were then added to the Falcone tube and incubated at room temperature for 20 min. F-DMEM was added to a final volume of 50 mL after the incubation. The C-DMEM was aspirated from the HEK293 cells and washed with F-DMEM prior to adding the transfection mixture. The F-DMEM mixture was then added, where each well on the plate contained a final volume of 2 mL, and incubated for 48 h at 37 °C. By the end of the incubation period, the cells typically changed color to orange, being careful not to over-incubate (resulting in yellow color). The supernatant was collected from the wells and placed into 50 mL falcon tubes. These tubes were centrifuged for 7 min at $290 \times g$ at 4 °C. Being careful not to disturb the bottom of the tubes, the supernatant was, once again, recovered and filtered through a 0.22 µm syringe filter. To ensure longevity of the samples, 1 mL aliquots were frozen and stored at −80 °C until needed for use. The yield

of the pseudoparticles was typically $10^9$–$10^{12}$ particles/mL and all stocks were adjusted to a final concentration of $10^9$ particles/mL.

## Pseudotyped Particle transduction (infectivity) assay

Spike-containing viral pseudoparticles ($VPPs_{Spike}$) were produced as mimics of SARS-CoV-2 infectious virions using previously established methodologies[22]. The backbone of the VPPs consisted of a Murine Leukemia Virus (MLV)-Gag-Pol, and the viral envelope contained wtSARS-CoV-2 Spike protein (WH1 strain), referred to as wt in the bar graphs here. The interior cavity of the particles contained a luciferase reporter gene, which allowed for a straightforward method to test the transduction of the $VPP_{Spike}$. In this assay, once the reporter gene was successfully delivered and integrated into the host cell's genome, the transduced cells were quantified using a luciferase activity assay. To perform this assay, African green monkey kidney epithelial Vero E6 cells were seeded in 24-well plates and incubated until 80–90% confluency was obtained. Each well was washed with 0.5 mLs of DPBS 3×, inoculated with 0.2 mLs of undiluted pseudovirus particle solution, and incubated at 37 °C for 1.5 h while agitating on a rocker. After the first incubation period was complete, 0.2 mLs of C-DMEM were added and incubated at 37 °C for 72 h. The infectivity was assessed using previously reported luciferase assay[22]. Briefly, the luciferase substrate and 5× Promega lysis buffer were thawed. The buffer was diluted with sterile water and added to the cells for lysis. For most effective lysis, the cells went through several freeze-thaw cycles, being transferred from −80 °C to room temperature 3×. After the last thaw cycle, 10 µL of lysate and 20 µL of Luciferin were combined in an Eppendorf tube and analyzed using a Promega (Durham, NC) GlowMax 20/20 luminometer. All experiments were repeated with a minimum of three biological replicates to ensure reproducibility.

## Transfection of plasmids containing SARS-CoV-2 Spike

Typically the SARS-CoV-2 Spike was transfected into HEK293 cells. For a 10 cm petri dish 400 µL of Opti-MEM was combined with 24 µL of Lipofectamine and incubated for 5 min at room temperature. In another tube, 8 µg of plasmid was added to 400 µL of Opti-MEM. The two tubes were combined and incubated further for 20 min at room temperature. Once the appropriate cells reached ~70% confluency, they were washed with DPBS. The Opti-MEM solution, containing transfection reagent and the plasmid, was added directly to the cells. The cells were incubated at 37 °C for 1 h, and then 8 mLs of C-DMEM were added to the top of the cells as well. They continued to incubate at 37 °C for 12–16 h before the next step.

## SLB formation on PEDOT:PSS surface

PEDOT:PSS coverslip/ electrode devices were soaked in DI for over 24 h prior to use. Cell blebs and POPC liposomes were mixed and sonicated for 20 mins to induce fusion[57,58] before adding onto an oxygen plasma-treated (Harrick Plasma Inc., Ithaca NY, PDC-32G, 7.2 W, 350 Micron, 1 min) PEDOT:PSS surface. It is worth noting that the plasma condition needs to be tuned for each plasma cleaner, as weak treatment won't provide sufficient hydrophilicity to rupture the blebs and vesicles, while too strong of a treatment will render the surface more negatively charged and rougher, making it challenging for the often negative native components to self-assemble into a mobile SLB. The incubation time for SLB formation on PEDOT:PSS surface was 1 h before excess materials were rinsed out with PBS buffer prior to further characterization. The presence of native membrane components in the SLB was observed using TIRF as shown in Fig. 2b and Supplementary Figs. 7–9.

## FRAP analysis

Cell blebs were sonicated for 30 min (kept under 25 °C with ice pad) to incorporate the lipophilic dye octadecyl rhodamine B chloride (R18) into the blebs (1 µL of 0.5 mg/mL R18 into 100 µL of blebs). SLB

formation proceeded as previously described. To verify formation of the SLB and confirm lipid mobility, an inverted Zeiss Axio Observer Z1 microscope was used with a ×20 objective lens. A 20 μm diameter was bleached for 500 ms and the recovery was monitored for 30 min. The fluorescence intensity was recorded and normalized. The data was fit to a standard Bessel function and diffusion (*D*) was determined using the equation: $D = w^2/4t_{1/2}$, where *w* represents the width (diameter) of the bleach spot, and $t_{1/2}$ is the time it took for the fluorescence to recover to half of the maximum intensity, and *D* is the determined diffusion measurement. All experiments were repeated with a minimum of three biological replicates.

## TIRF microscopy

SLBs were prepared (without the R18 dye as previously described). The pseudoparticles were first labeled by sonicating with R18 dye (1 μL of 0.5 mg/mL for 100 μL of pseudo particle solution) for 30 min (kept under 25 °C with ice pad). For this assay, the VPPs are labeled to a semi-quenched state (independently verified using a fluorimeter (Supplementary Fig. 11)), where the fluorescence intensity is adequate to observe the particles within the TIRF FOV but not proportional to the extent of labeling. The excess dye was removed using a size-exclusion column or simply washed away when appropriate. TIRF measurements were performed on Zeiss Axio Observer.Z1 microscope using an α Plan-Apochromat ×100 objective with a numerical aperture (NA) of 1.46. The samples were excited with a 561 nm laser and the angle of incidence was adjusted to ~68° to insure an evanescent wave of 100 nm with total internal reflection. Prior to acquiring these images, we washed our experimental well with excessive buffer to remove any unbound particles and ensure that we were acquiring images of only those particles that were bound and not diffusing in/out of the FOV.

## Microelectrode fabrication

Gold contact pads were patterned on fused silica wafers using a standard photolithography procedure: exposure, develop, deposition, and lift-off[59]. A 200 nm of $SiO_2$ insulating layer was then deposited ubiquitously on the Au patterned wafer using plasma enhanced chemical vapor deposition (PECVD). A second layer of photolithography was applied to define the PEDOT:PSS electrode locations on the gold contact pad, followed by the reactive ion etching of $SiO_2$ until it reached the gold surface. PEDOT:PSS mixed with 1 v/v% of GOPS was then spin-coated at 4000 rpm on both exposed gold contact and $SiO_2$ insulating layer, followed by annealing at 140 °C for 30 min to drive off all water. A third layer of photolithography was applied to remove the PEDOT:PSS spun on $SiO_2$, taking advantage of the germanium (Ge) hard mask protocol previously reported[60]. The 100 nm thick protective Ge hard mask on PEDOT:PSS electrode was then removed by immersing in deionized water for 48 h. It is worth mentioning no other additives, such as ethylene glycol, were added to improve the conductivity of PEDOT:PSS film, hence the PEDOT:PSS electrode baseline varies from PEDOT:PSS electrode baseline with ethylene glycol as shown in Supplementary Fig 12. The acidic PBS buffer (pH 5.5) used to activate CatL for late entry showed no impact on electrode baseline (Supplementary Fig. 12).

## EIS measurement and data analysis

An Autolab PGSTAT302N potentiostat was used to conduct the EIS measurements. The frequency of applied sinusoidal voltage was swept from $10^6$ Hz to 1 Hz to capture the change in electrical signal at each step after the addition of biological materials. Prior to SLB formation, the PEDOT:PSS electrode baseline was measured and fit to a RC circuit. Signals after SLB formation were fit to a RC(RC) circuit using NOVA software package (Metrohm AG), where membrane resistance and capacitance were extracted and then normalized against electrode area (a summary of fitted values for Figs. 3–5 are included in

Supplementary Table 1). Copious rinsing with PBS buffer was done before each measurement at every step after SLB formation, 20 mins after adding VPPs (5 μL of 4× diluted VPPs stock, ~$10^6$ particles) or cell blebs (90 μL, ~$10^8$ blebs) and 30 mins after adding CatL. All experiments were repeated with a minimum of three biological replicates.

## Optimization of conditions for fusion assays

Prior to SLB formation, Vero E6, TMPRSS enhanced Vero E6, and HEK-293T cells lines were verified for the presence of appropriate proteins (Supplementary Fig 1). To ensure that there was minimal variability between cell batches, we used Western Blot analysis and ran three biological replicates that showed consistent presence of either ACE2 in Vero E6 and TMPRSS2 enhanced Vero E6 and TMPRSS2 in TMPRSS2 enhanced Vero E6 cells. Blebs containing these proteins were then used for SLB formation using the protocol outlined above.

The VPP concentrations used for these experiments were conducted such that VPPs were always in excess to ensure that VPP-ACE2 receptor interactions were maximized. Doing so ensured the most detectable change in signal and any changes in signal would likely be a result of varying fusogenicity between the variants. For the early pathway, the SLB was incubated in Reaction Buffer A and VPPs were added to trigger fusion. For experiments using TMPRSS2 inhibitor, we prepared 10 mM stock solutions of camostate mesylate from MedChemExpress (1 Deer Park Dr, Monmouth Junction, NJ 08852) in DMSO. The final concentration of camostat mesylate added to Reaction Buffer A was 50 μM. For the late entry pathway, we used commercially available cathepsin L from R&D Systems (614 McKinley Pl NE, Minneapolis, MN 55413) provided in 16.9 μM solution (50 mM sodium acetate 500 mM NaCl, pH 5.0). This stock was diluted using Reaction Buffer B to 169 nM and the final concentration of CatL in the reaction solution was 1.7 nM for the optical assay and 10 nM for the electrical assay. The final concentration of VPPs in all experiments was kept at $10^7$ particles/mL.

## Data and statistical analysis

The number (*n*) of replicates used per group is described in each figure legend and represents technical replicates. All fusion experiments plotted in Fig. 3 were performed with three biological replicates. Data distribution analyses were performed with Prism. Ordinary one-way ANOVA with multiple comparisons with Šídák's or Tukey's multiple comparisons tests was used to evaluate experiments containing multiple groups as noted in the legends of Figs. 3 and 6. Unpaired two-sided *t* test was used to compare differences between the two groups as indicated in the legend of Fig. 4. The upper threshold for statistical significance for all experiments was set at $p < 0.05$.

## Reporting summary

Further information on research design is available in the Nature Portfolio Reporting Summary linked to this article.

# Data availability

All data supporting the findings of this study are available within the article and its supplementary files. Any additional requests for information can be directed to and will be fulfilled by the corresponding authors. Source data are provided with this paper.

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

## Acknowledgements
S.D. and R.O. acknowledge funding for this project, sponsored by the Defense Advanced Research Projects Agency (DARPA) Army Research Office and accomplished under Cooperative Agreement Number W911NF-18-2-0152. The views and conclusions contained in this document are those of the authors and should not be interpreted as representing the official policies, either expressed or implied, of DARPA, the Army Research Office or the U.S. Government. The U.S. Government is authorized to reproduce and distribute reprints for Government purposes, notwithstanding any copyright notation herein. The fabrication of microelectrodes in this work was performed at the Cornell NanoScale Facility, a member of the National Nanotechnology Coordinated Infrastructure (NNCI), which is supported by the National Science Foundation (Grant NNCI-2025233). Z.C. and S.D. acknowledge the Smith Fellowship for Postdoctoral Innovation from Cornell University. Z. C. acknowledges the Eric and Wendy Schmidt AI in Science Postdoctoral Fellowship, a Schmidt Futures program. A.P. acknowledges support from the National Science Foundation Graduate Research Fellowship Program. We thank Juliana D. Carten and Jordan P. Fitzgerald for useful discussions and assistance with the editing of the final manuscript. Figures 1–3 and Supplementary Figs. S7 and S9 were partially created with BioRender.com with a license.

## Author contributions
Conceptualization: Zhongmou Chao, Ekaterina Selivanovitch, Konstantinos Kallitsis, Zixuan Lu, Róisín Owens, Susan Daniel. Methodology and visualization: Zhongmou Chao, Ekaterina Selivanovitch. Investigation: Zhongmou Chao, Ekaterina Selivanovitch, Ambika Pachaury. Supervision: Róisín Owens, Susan Daniel. Writing—original draft: Zhongmou Chao, Ekaterina Selivanovitch, Susan Daniel. Writing—review & editing: Zhongmou Chao, Ekaterina Selivanovitch, Konstantinos Kallitsis, Zixuan Lu, Ambika Pachaury, Róisín Owens, Susan Daniel.

## Competing interests
The authors declare no competing interests.
