## [Peer Review File · Nature Communications]

REVIEWER COMMENTS

Reviewer #1 (Remarks to the Author):

The authors described an electronic biomembrane device that could be used to recapitulate and detect viral entry in a cell-free way by using SARS-CoV-2 as a model. Although this method seems promising, this reviewer has a couple of concerns and comments as below.

1. In addition to the components used in the present study, viral entry of SARS-CoV-2 under physiological conditions involves more complex processes, such as furin-mediated cleavage of spike (S) protein and clathrin/caveolin-mediated endocytosis. The use of the present approach could be limited to mimicking and detecting S protein-mediated receptor binding and cell-virus membrane fusion. Therefore, the application scope of this approach should be narrowed.

2. A major problem is that the manuscript lacks detailed and accurate parameter information, including the dosages of viral particles used in each experiment, the amount of S protein in different VPPs, the amount of ACE2 on membrane, the amount of TMPRSS2 on membrane, the amount of CatL in this system, and the amount of S protein on membrane, etc. These parameters are critical for establishment and demonstration of a stable viral infection-on-a-biomembrane chip platform that can be reproduced and accessible by other researchers' lab and should be provided in the text or figure legends.

3. A comparison between S-containing viral pseudoparticles' and authentic viruses' infection on the device is suggested to demonstrate the feasibility of this model in reflecting natural infection.

4. Why the curves and values of +Vero SLB (+TMPRSS2) in Fig. S7g and Fig. S7i are significantly different? Don't they use consistent experimental parameters?

5. Given the device could replicate partial processes of SARS-CoV-2 viral entry into host cells, a more accurate title should be considered, such as "An electronic biomembrane device for cell-free sensing entry of SARS-CoV-2 viruses", "An electronic biomembrane chip for cell-free recreation of SARS-CoV-2 viral entry", or "An electronic biomembrane chip for cell-free recreation of SARS-CoV-2 Spike protein-mediated receptor binding and membrane fusion".

6. Experimental evidence are needed to support the statements that this infection-on-chip platform can be used to screen antibodies against Spike protein and small molecule fusion inhibitors in a high throughput manner.

7. In line 252, word is missing between ACE2 and TMPRSS2.

8. In line 516, reference is missing.

Reviewer #2 (Remarks to the Author):

The article by Zhongmou Chao et al. entitled “Recreating the biological steps of viral infection on a bioelectronic platform to profile viral variants of concern” present an electronic biosensing chip capable of probing both virus binding and fusion to a membrane. The platform consists of a supported lipid bilayer prepared from cell blebs and thus containing native cell membrane components to probe virus binding and fusion. Signal transduction relies on electrochemical impedance spectroscopy to measure changes in electrical properties of the bilayer. The article demonstrates that both virus binding and fusion lead to detectable electrical signal changes on the chip. The authors demonstrate that the platform can be used to recapitulate different entry pathways and to probe the fusogenic potential of one type of spike to different cell types, by reversing the assay (i.e. creating a bilayer containing spike and adding different bleb types). They also show that the platform can detect changes in fusogenic potential between different SARS-COV-2 variants of concerns.

The greatest novelty of this work lies with the combination of the native membrane platform with electric read out of binding and fusion events which potentially allows for the preparation of integrated biosensing chips for rapid assessment of fusogenic potential of virus-containing samples. This may be useful in the context of assessing the fusogenicity of various VOC or of assessing the amount of entry-competent virus particles in a sample, which eventually would represent a progress over diagnostic assays based on detecting viral proteins or genetic material.

In principle, such experiments can be performed by fluorescence (as demonstrated by control experiments presented in this manuscript and by previous work by both these authors and others). However, the infection-on-chip device offers the additional possibility of doing such experiments label free.

Overall, the manuscript is likely to be of interest to the readership of nature communication. However, I recommend that the following comments are addressed before publication.

Comments:

1)

Figure 2B presents a control experiment where the binding of VPP particles containing spike is compared to the binding of VPP which do not contain the viral protein. While this experiment does indeed prove that binding to the bilayer is specific to the presence of native membrane material, I do not agree that it clearly proves that this reflects binding to ACE2. In principle the virus could also be binding to other molecules in the bilayer. Along these lines I would suggest to rephrase sentences such as

“visualization of binding events between ACE2 and VPPWH1 was necessary to verify that native cell receptors from blebs were incorporated into the SLB” line 203-205

“specifically bound to ACE2 assembled SLB” line 210

Or even “ the presence of native membrane components in SLB was verified using TIRF as shown in Figure 2B. line 543; as such experiments are only an indirect proof for the presence of native components.

Part of the beauty of the bleb-derived supported lipid bilayer system that is used is that it has all the components of the competent cell lines present and thus the identity of all known interaction partners is not required in order to observe binding and fusion. If the authors really want to be sure ACE2 is the key binding component in the above statements, then perhaps an antibody-blocking experiment would best show the specific role of ACE2.

2)

I could not find any detailed information on how membrane resistance is calculated. This should be included / detailed more clearly in the methods section.

3)

The authors should clarify what is meant with fusogenic potential / fusogenicity. Presumably the percentage of particles that fuse? Or does it have to do with the fusion kinetics of the particles?

4)

I could not find a clear description of how the fusion experiments are performed, with respect to the concentration of particles used. When comparing e.g. the fusogenic potential of two VOCs, it is presumably important that the same amount of virus particles is added in both cases? Or is this not the case? Related: is the electrical signal (i.e., shift in bilayer resistance) dependent on the amount of virus particles which fuse with the membrane?

5)

It is not clear to me how one can unambiguously distinguish binding from fusion without help of microscopy images. The authors should provide more clear guidelines concerning this. I understand that the ΔR values are different. But are such values concentration independent? For example: Could many particles binding be interpreted as few particles fusing or vice-versa?

6)

lines 176-177, “The particles count, provided by NTA analysis, allowed us to control the relative concentrations of the VPPWH1 and blebs used to assemble the SLBs.” I believe there is a typo, “..VPPWH1 and blebs...” should perhaps say “...blebs and POPC vesicles...”? The VPPWH1

particles are not use to assemble SLBs... Also, the authors should provide more details on how they produce SLBs reproducibly with the same amount (%) of bleb material from the different bleb sources. The above sentence is all that is said on the subject and this control over SLB composition seems key to experiment reproducibility.

7)

lines 144-145, “The fusogenic vesicles used in this work are reconstituted from purified 1-oleoyl-2-palmitoyl-sn-glycero-3-phosphocholine (POPC) lipids, which is the principal lipid component of mammalian and viral membranes.” POPC lipids from Avanti are synthetic, not a purified natural product. While Avanti does mention that POPC is the most abundant lipid in their Egg PC extract, I believe the truth is that “oleoyl” and “palmitoyl” are the most abundant of the fatty acid chains and PC is the most abundant headgroup in most lipidomic studies I have seen of mammalian membranes, but never have I seen the intact POPC reported. Also, viral envelopes are derived from mammalian membranes and thus the lipid abundances are fairly similar. In short, this statement is a bit too “matter-of-fact” when it is more of a simplified extrapolation of observations. If I am wrong, I would love to see the citation(s) that supports the statement “POPC is the principal lipid component of mammalian and viral membranes”.

Reviewer #3 (Remarks to the Author):

The manuscript entitled “Recreating the Biological Steps of Viral Infection on a Bioelectronic Platform to Profile Viral Variants of Concern” co-authored by Chao et al. describes a novel bioelectronics approach to study virus binding to the host cell membrane and subsequent fusion or endocytosis without using cells but membrane preparations instead. The authors have used a very timely example: SARS-CoV2 and variants. The authors reproduce for the known variants their individual fusogenic capacity and the mode of cell entry. The manuscript is thoroughly and clearly written, the scientific arguments are sound and the conclusions are strictly based on the experimental data. The manuscript will for sure find a lot of interest among the readership. This “infection on chip” approach provides a new and very practical new technology to perform very detailed studies on new virus species, mutants and mechanisms of infection. The reviewer is particularly impressed that the technology would not require genetically engineered cells DURING the assay – very different from most cell-based assays. Work with GMOs is only required to prepare the sensors but not for the final assays. This pave the way for a widespread use. Using membrane resistance as an indicator of viral fusion and assuming that fusion leads to an increase in resistance by a yet unknown mechanism, the data is without contradiction. Solely a clear explanation why R_m increases upon fusion and other questions regarding the electrical parameters of the membrane (see below) require more explanation and revision. I strongly recommend this manuscript for publication after my major and minor concerns have been addressed.

Major concerns:

Line 224: The measured spectrum of the SLB-free PEDOT electrodes differs from the simplifying illustration in the center of fig 2C. The experimental spectrum shows a minor dispersion between 1 kHz and 10 kHz so that it is not just a resistor-capacitor in series structure as the authors claim in line 225. Please phrase more cautiously in this respect. What is the origin of the central dispersion? It must have to do with the PEDOT/PSS on the electrode. The reviewer is aware that this not in the center of this manuscript and certainly does not interfere with the story. But this dispersion is not due to any biological component and it is just masked upon coating the PEDOT-electrode by an SLB. It is still there and contributes to the readings. So it may even change upon acidification. What is the impact of the pH drop on PEDOT/PSS conductivity? Have the authors checked for the response of a pure PEDOT/PSS coated electrode upon pH shift? The reviewer understands that the PEDOT is covered and shielded by the SLB but is this truly complete?

The authors present an equivalent circuit that contains the membrane resistance R_m and the capacitance C_m but the values for C_m are not reported at all. Why? Values of C_m are very potent indicators for the quality of the membrane preparation so it must be reported.

Figure 2C: According to figure 2C, the resistance of the SLB is $12.3 \text{ Ohm} \cdot \text{cm}^2$. And it is in the same order of magnitude in the other examples. This is very low and about two orders of magnitude lower than expected for lipid bilayers without any ion channels. Why is the specific membrane resistance so low? Is the SLB covering the electrode completely? Since (i) there is considerable scattering in R_m of different SLB preparations and (ii) R_m values are too low, it seems plausible that coverage is not complete, that there are gaps affecting the R_m readings. Any indication? The authors have to comment on the completeness of coverage. It is hard to judge in particular as the authors do not provide the electrode surface area – unless it escaped my attention.

Line 246: The increase in resistance upon viral fusion is unexpected and counter-intuitive. The membrane surface area increases upon fusion of SLB and virus membrane so that the area-specific resistance should decrease. The authors speculate that additional biomacromolecules from the viral membrane get transferred into the SLB or gaps are filled (see above). To the reviewer's understanding, such a transfer of biomolecules should lead to an increased membrane capacitance C_m . A protein-free bilayer has about $0.7 \mu\text{F}/\text{cm}^2$ whereas mammalian cell membranes with regular protein content show app. $1 \mu\text{F}/\text{cm}^2$ due to the presence of protein with higher dielectric constant. It would support the discussion and help the conclusion to report the C_m values next to R_m values. Also: A rather simple experiment may lead to more insight and I am almost sure that the authors have done it: titration experiments with sequentially increasing virus load on one and the same SLB. The more VPPs fuse, the more the electrical signals should change.

Why are the electrical resistances of SLBs with identical composition so different from experiment to experiment? For example: Vero SLB + ACE2 + TMRSS2 is 18.7 Ohm*cm² or 11.3 Ohm*cm². Is the SLB preparation not yielding complete membrane coverage of the electrode? Are the electrode edges completely covered?

Minor Concerns:

Line 177: Supposedly „concentrations of the VPPwh” should read “concentration of vesicles”.

Line 223: Please reword “.. the circuital response to alternating voltage with changing frequency...” maybe like “frequency-dependent impedance....”.

Line 259: The data in figure 3b suggest that the resistance increase due to late entry is significantly bigger compared to the early entry. The data pooled in figure 3c indicate the opposite: late stage entry increases the resistance less than early state entry. Isn't this contradictory?

Line 354: Rephrase „surface elctrodynamics measurement“ to maybe “dielectric measurements”.

Response to reviewers' comments

Reviewer #1:

The authors described an electronic biomembrane device that could be used to recapitulate and detect viral entry in a cell-free way by using SARS-CoV-2 as a model. Although this method seems promising, this reviewer has a couple of concerns and comments as below.

1. In addition to the components used in the present study, viral entry of SARS-CoV-2 under physiological conditions involves more complex processes, such as furin-mediated cleavage of spike (S) protein and clathrin/caveolin-mediated endocytosis. The use of the present approach could be limited to mimicking and detecting S protein-mediated receptor binding and cell-virus membrane fusion. Therefore, the application scope of this approach should be narrowed.

We thank the reviewer for the feedback and insightful comments on our work. We agree that we are only capturing the initial part of the infection process, viral entry, namely binding and fusion. However, other aspects of the platform that are novel include the ability to detect active, whole viruses capable of fusion via one of the two known major biological pathways- 1) the early entry pathway, in which fusion takes place at or close to the cell membrane after the S2 Spike domain is cleaved via TMPRSS2, and 2) the late entry pathway, in which the particles are first endocytosed, and then cleaved at the S2' site by cathepsins to trigger fusion. We understand that endocytic mechanisms may involve caveolae-dependent/clathrin-independent, clathrin-dependent, and caveolae and clathrin-independent steps. Our platform does not mimic the actual endocytosis process itself; instead, we re-create the chemical cues inside the endosome that lead to the triggering of the fusion of the viral envelop with the endosome membrane and thereby mimic the local endosome conditions in our platform that induce fusion in this cellular compartment. The critical cues are established to be low pH activation of cathepsin L cleavage of the S2' site to trigger fusion peptide release and commencement of the membrane fusion process. The reviewer is correct that a prior cleavage event is necessary to 'prime' the virus at the S1/S2 site where furin and other proteases have been shown to be critical. Priming of particles by furin or furin-like proteases can certainly be built into the protocol of this device's operation, but the particles that were used in this study contain Spike proteins that have already been primed at the S1/S2 sites. To avoid confusion for our readers, we have added a few sentences in the manuscript to highlight the potential different pathways by which endocytosis can take place and clarified how we are mimicking the fusion conditions of the endocytosis pathway. The main developments we show in this assay is the ability to re-create the biological pathways that lead to fusion and distinguish the relative fusogenicity of virus particles/spikes across variants.

We are unclear if the reviewer is also concerned that this platform may only apply to coronavirus and therefore be of limited value. The platform does work for other membrane-enveloped viruses with other fusion triggering mechanisms. We have recently achieved success with the measles virus (data can be supplied for review by

request) and have had success with influenza as well. As such, we believe this platform can be used to study other viruses when the appropriate triggers are integrated into the experimental execution.

2. A major problem is that the manuscript lacks detailed and accurate parameter information, including the dosages of viral particles used in each experiment, the amount of S protein in different VPPs, the amount of ACE2 on membrane, the amount of TMPRSS2 on membrane, the amount of CatL in this system, and the amount of S protein on membrane, etc. These parameters are critical for establishment and demonstration of a stable viral infection-on-a-biomembrane chip platform that can be reproduced and accessible by other researchers' lab and should be provided in the text or figure legends.

Thank you for pointing this out as we want to ensure our assay is accessible to all researchers. We have added the pseudoparticle and cathepsin L concentrations used for our experiments. Additionally, we have added a new section to the Methods providing experimental details for the fusion conditions used. We did not tag ACE2, TMPRSS2, and Spike protein for isolations because we wanted to maintain their native integrity. To verify that we were using comparable amounts of proteins across our experiments, we used semi-quantitative western blotting analysis to show minimal variability in protein expression in the case of ACES2 and TMPRSS2 and transfection efficiency in the case of Spike protein. We have added Fig S1 to our manuscript to share these data. Though it is indeed difficult to specify exactly the same amount of say the receptor or protease in the blebs and SLBs created from batch to batch, we note that we have qualitatively observed that when we have more receptors, we see more fusion, and when we have less, we observe less fusion (and binding). Thus we are confident that other labs should be able to reproduce this platform, even with some variation in concentrations, and would be able to compare results to their internal baseline to obtain similar normalized outcomes to what we present here. In fact, similar experiments have been carried out in the Owens lab with completely different virus stocks, bilayer formulations, and materials yielding similar results (forthcoming publication).

3. A comparison between S-containing viral pseudoparticles' and authentic viruses' infection on the device is suggested to demonstrate the feasibility of this model in reflecting natural infection.

We agree with the reviewer that one of the potential applications for our devices can be for the detection of live viruses. However, the requirement for a BSL-3 facility outfitted with the required electrical equipment hinders our ability to perform these experiments at this time. Based on extensive research and numerous publications on this topic, pseudo-typed particles or pseudovirions have successfully been demonstrated as models for authentic viruses and only require a BSL-2 facility, which we currently have in our lab space. We have added a section to our manuscript that acknowledges this and provides citations as examples of studies completed with pseudoparticles as viable model systems to represent the native particle entry behaviors.

4. Why the curves and values of +Vero SLB (+TMPRSS2) in Fig. S7g and Fig. S7i are significantly different? Don't they use consistent experimental parameters?

One challenge of self-assembling SLB via vesicle fusion on electrodes coated with the conducting polymer is the variation in impedance that results from variation in PEDOT.PSS coating, but more so from the variation in self-assembly of the supported bilayer on it. We endeavored to keep experimental conditions as consistent as possible. Nonetheless, we do see variation in the absolute magnitudes of the impedance. After many years of working with these biomembrane devices, we conclude that the magnitude of SLB impedance signal is a reflection of imperfections in the lipid bilayer formed on the electrode, and it also depends on such factors such as composition (i.e. amount of blebbed material), plasma treatment condition of the PEDOT.PSS prior to SLB coating, and device configurations (electrode dimension and features). Over the years our lab has made significant progress in tuning SLB formation conditions on PEDOT.PSS, and has gotten a better handle on the variability, reflected in the figure below (Figure I). This histogram of the distribution of resistance values across multiple devices and various

electrode dimensions (a total of 56 experiments in total) is for hybrid SLBs (made from HEK blebs ruptured by POPC). While this variation may be non-ideal, it does reveal that the results we obtain are consistent despite this variation when the measurements are normalized to their internal baseline. By comparing the differences between the baseline pre- and post-treatment, we can determine the relative change when the system is altered in some way, such as during membrane fusion.

While resistance values are more sensitive to imperfections in the final SLB, the capacitance value provides us with a good confirmation of bilayer formation. If we consider the bilayer as a capacitor of two plates separated by a dielectric material, we expect based on the dimensions of the bilayer thickness (~ 4-5 nm) that capacitance will roughly fall in the range of $1 \mu\text{F}/\text{cm}^2$. Capacitance becomes a good sanity check for bilayer formation because adsorbed, unruptured vesicles (on the order of 100 nm diameter) result in capacitance values far different from the SLB value. Finally, optical data also provides a secondary independent verification of bilayer formation. Unruptured vesicles cannot recover from photobleaching, but SLBs do, as seen in our FRAP analysis.

In conclusion, while the resultant impedance values of the membrane-coated sensor can vary from device to device, the ability to carry out repeatable measurements is less impacted because we are always examining

changes relative to the starting baseline value.

Figure I. A histogram of normalized resistance of SLBs recapitulated from HEK cell blebs using POPC as fusogenic lipids. The x-axis is the resistance value from the fit of the impedance data for each experiment normalized by electrode area. The y-axis is the count number.

5. Given the device could replicate partial processes of SARS-CoV-2 viral entry into host cells, a more accurate title should be considered, such as “An electronic biomembrane device for cell-free sensing entry of SARS-CoV-2 viruses”, “An electronic biomembrane chip for cell-free recreation of SARS-CoV-2 viral entry”, or “An electronic biomembrane chip for cell-free recreation of SARS-CoV-2 Spike protein-mediated receptor binding and membrane fusion”.

We agree that we should be clear that this work is for a cell-free system and have thus changed the title to "Recreating the Viral Infection on Bioelectronic Viral Variants of like to highlight in provides an VOCs and opted title intact.

6. Experimental support the infection-on-chip screen antibodies and small inhibitors in a manner.

This is an and is exactly the In ongoing work, inhibitors as and use this binding and support our claim

Biological Steps of a Cell-free Platform to Profile Concern". We would our title that this work ability to distinguish to keep that part of the

evidence are needed to statements that this platform can be used to against Spike protein molecule fusion high throughput

excellent suggestion topic of our next paper. we used ACE2 potential therapeutics platform to determine fusion inhibition. To in this paper, we added

a different demonstration of screening a TMPRSS2 inhibitor (we use known inhibitor camostat mesylate) and show that indeed fusion function can be suppressed upon its addition to the reaction milieu. The data for these experiments can be found in the figure below (Figure II) and this new section has been added to the manuscript.

Experiment details: following a method reported in the Nature paper,¹ we explored several different approaches of introducing camostat mesylate into our system (since this is cell-free): 1) 50 μ M of camostat mesylate was added on top of an hybrid SLB made from Vero (+ TMPRSS2) and incubated for 2h; 2) 50 μ M of camostat mesylate was added to freshly harvested Vero blebs (+ TMPRSS2) and incubated in the refrigerator overnight prior to hybrid SLB formation; 3) 50 μ M of camostat mesylate was added to freshly harvested Vero blebs (+ TMPRSS2) and POPC liposomes, mixed, and sonicated prior to hybrid SLB formation. For all these cases, after SLB formation, we then added VPPs (Wuhan-Hu-1) and collected the electrical responses. For the positive control, a no camostat mesylate case was tested as a comparison.

Figure II. Impedance change of Vero SLBs (+ TMPRSS2) caused by fusion with VPPs via early entry pathway under different treatments using camostat mesylate (50 μ M) as the TMPRSS2 inhibitor. Control group= no camostat mesylate added, experiment 1= incubate SLB with camostat mesylate, experiment 2= cell blebs incubate with camostat mesylate overnight prior to SLB formation, experiment 3= sonicate cell blebs with camostat mesylate prior to SLB formation.

As shown in Figure II, 50 μ M of camostat mesylate significantly suppressed SLB impedance change from fusion with VPPs for all three experimental groups compared to the control group, with camostat mesylate incubating with vero SLB (+TMPRSS2) being the most effective in blocking fusion with Spike VPPs. It is worth mentioning the control group here mirrors fusion events via early entry pathway described in the manuscript, though it has a significantly higher impedance increase (+ 147% vs. + 54%). Initially we were concerned that this reflected a much larger change than observed with the previous batch of particles. However, upon testing of this batch of VPPs with the cell-based luciferase assay, these particles were significantly more active compared to the prior batches as shown in luciferase assay below (almost a half log), Figure. III. Though we did not intend to test two batches that vary significantly in activity, we nonetheless discover here that our device response aligns with the biological assay once more. Finally, this also further corroborates the point made above that batch-to-batch variability (in device, SLB, and here in the VPP activity) will not change the ability of the device to assess viral entry when using the baseline as a reference point.

Figure III. Relative transduction efficiencies of the Wuhan-Hu-1 Spike pseudoparticles used in the manuscript (wt Batch 1) and in this study (wt Batch 2).

7. In line 252, word is missing between ACE2 and TMPRSS2.

This has been corrected.

8. In line 516, reference is missing.

Thank you, we have added the appropriate reference.

Reviewer #2:

The article by Zhongmou Chao et al. entitled “Recreating the biological steps of viral infection on a bioelectronic platform to profile viral variants of concern” present an electronic biosensing chip capable of probing both virus binding and fusion to a membrane. The platform consists of a supported lipid bilayer prepared from cell blebs and thus containing native cell membrane components to probe virus binding and fusion. Signal transduction relies on electrochemical impedance spectroscopy to measure changes in electrical properties of the bilayer. The article demonstrates that both virus binding and fusion lead to detectable electrical signal changes on the chip. The authors demonstrate that the platform can be used to recapitulate different entry pathways and to probe the fusogenic potential of one type of spike to different cell types, by reversing the assay (i.e. creating a bilayer containing spike and adding different bleb types). They also show that the platform can detect changes in fusogenic potential between different SARS-COV-2 variants of concerns.

The greatest novelty of this work lies with the combination of the native membrane platform with electric read out of binding and fusion events which potentially allows for the preparation of integrated biosensing chips for rapid assessment of fusogenic potential of virus-containing samples. This may be useful in the context of assessing the fusogenicity of various VOC or of assessing the amount of entry-competent virus particles in a sample, which eventually would represent a progress over diagnostic assays based on detecting viral proteins or genetic material. In principle, such experiments can be performed by fluorescence (as demonstrated by control experiments presented in this manuscript and by previous work by both these authors and others). However, the infection-on-chip device offers the additional possibility of doing such experiments label free.

Overall, the manuscript is likely to be of interest to the readership of nature communication. However, I recommend that the following comments are addressed before publication.

Comments:

1) Figure 2B presents a control experiment where the binding of VPP particles containing spike is compared to the binding of VPP which do not contain the viral protein. While this experiment does indeed prove that binding to the bilayer is specific to the presence of native membrane material, I do not agree that it clearly proves that this reflects binding to ACE2. In principle the virus could also be binding to other molecules in the bilayer. Along these lines I would suggest to rephrase sentences such as:

“visualization of binding events between ACE2 and VPPWH1 was necessary to verify that native cell receptors from blebs were incorporated into the SLB” line 203-205

“specifically bound to ACE2 assembled SLB” line 210

Or even “ the presence of native membrane components in SLB was verified using TIRF as shown in Figure 2B. line 543; as such experiments are only an indirect proof for the presence of native components.

Part of the beauty of the bleb-derived supported lipid bilayer system that is used is that it has all the components of the competent cell lines present and thus the identity of all known interaction partners is not required in order to observe binding and fusion. If the authors really want to be sure ACE2 is the key binding component in the above statements, then perhaps an antibody-blocking experiment would best show the specific role of ACE2.

We thank the reviewer for the feedback and agree that the initially presented data did not verify a specific interaction between the pseudoparticle's spike protein and the ACE2 receptor located on the SLB. To further verify the specificity of this interaction, we prepared a SLB, as described in the manuscript, without the ACE2

receptor using a HEK 293T bleb-derived hybrid bilayer and saw that few binding events were observed. These data support a specific binding interaction between Spike protein and ACE2 receptor in our device. We have added this new data to the Supplemental Fig 8, provided a reference to the figure within the main text, and adjusted the language to reflect the reviewer's suggestions. We note for the purpose of this review that in an upcoming manuscript that elaborates on the antibody screening application of this work, antibody screening experiments, as suggested by the reviewer, do indeed show this specificity.

2) I could not find any detailed information on how membrane resistance is calculated. This should be included / detailed more clearly in the methods section.

A new section in Methods has been added to describe how we fit the EIS data and calculated normalized resistance and capacitance of the SLBs on these devices. We provide the circuit model and the process to analyze the data that leads to the extraction of R and C of the SLB, in addition to the R of the buffer and the C of the electrode.

3) The authors should clarify what is meant with fusogenic potential/fusogenicity. Presumably the percentage of particles that fuse? Or does it have to do with the fusion kinetics of the particles?

We agree this was unclear. We have added clarifying statements to the manuscript distinguishing between fusogenic vesicles described early in the Results section and fusogenic pseudoparticles described in later sections. The reviewer is correct in how we define these terms. Fusogenicity (of the pseudoparticles) is used to describe the relative number of particles that can fuse their membranes with that of the host membrane. "Fusogenic vesicles" refers to the liposomes we make that have a higher propensity to rupture and self-assemble into a planar, supported lipid bilayer. We agree that this terminology is confusing here given that we are discussing membrane fusion, but the supported bilayer community uses the terms "vesicle fusion" to refer to this SLB assembly process and so "fusogenic vesicles" has arisen as the way to describe vesicles that promote this process. We hope we have clarified this well enough in the manuscript now.

4) I could not find a clear description of how the fusion experiments are performed, with respect to the concentration of particles used. When comparing e.g. the fusogenic potential of two VOCs, it is presumably important that the same amount of virus particles is added in both cases? Or is this not the case? Related: is the electrical signal (i.e., shift in bilayer resistance) dependent on the amount of virus particles which fuse with the membrane?

Thank you for the comment, we have included a more detailed experimental description in the Methods section.

Yes, when comparing the fusogenic potential of different VOCs, the amount of virus particles we added was kept constant by adjusting the particle concentration according to particle counts measured by nano tracking analysis (Nanosight). We also kept the receptor number in the hybrid SLBs constant to the best of our abilities by maintaining the same ratios of the cell blebs and POPC vesicles used to form SLB. Our western blots show that the expression levels of the ACE2 and TMPRSS2 are generally consistent from batch to batch, although we do realize that western blotting is semi-quantitative at best. Nonetheless, we endeavored to keep conditions as consistent as possible across experiments so that as fair a comparison could be made across different strains and variants.

The question of dose dependency is a good one and it is worth elaborating on this point here. There will be a dose response in the electrical readout (resistance) when the limiting factor in the experiment is the amount of virus available to bind receptors. However, in this work we are at the other end of the spectrum where virus particles are in excess. The rationale for operating in this regime is the following. If we are to compare the fusogenicity of various strains, defined as the relative number of particles that can fuse their membranes with that of the host

membrane, this value will depend on the receptor density, the virus particle concentration, and the fusion activity of the particles/spikes. We want to capture this last feature, because what we aim to report is the differences between VOCs that arise not from variation in receptor or particle count, but due to some aspect of the spike or particle itself that is defined by that variant. As such, the value of the change in resistance is a measure of the fusogenicity of that strain relative to other strains when the receptor count and particle counts are kept the same between experiments. A choice then has to be made as to what particle concentration to use. Indeed, any concentration could be selected, even one that is less than needed to bind all available receptors. However, to operate our sensor in such a way that maximizes the signal we obtain, all available receptors should be occupied to maximize the signal from fusion possible. In this work, we operate in an excess of pseudoparticles.

To ensure our concentration of particles was indeed in excess, we verified this in the following way. We conducted a dose response across several concentrations using a hybrid Vero SLB (+ TMRSS2), see Figure IV. Here, we cover a range from 2.5 uL to 10 uL of particles, and the resultant change in the impedance. This experiment was done by adding in 2.5 uL to start, sweeping through the frequency range and then proceeding to the next amount (5 uL, double the amount) and taking the frequency sweep again. This was carried out in this manner for 7.5 uL and finally, 10 uL. What we see is that with each addition (no rinsing between) the resistance shifts upward. Initially we thought this was an indication of additional particle fusing (so a dose response) but when we rinsed the bilayer to remove excess particles after the 10 uL reading, we see the final data falls nearly back onto the 2.5 uL reading. This tells us that the amount of fused virus particles was already maximized after adding 2.5 L of particles and that the addition of more particles did not result in more fusion. This outcome makes sense if all the receptors that could capture virus were occupied already and so the addition of more viruses did not lead to any more viruses fusing with the membrane. For the experiments conducted for this manuscript, we chose to maximize the fusion signal by saturating the receptors/proteases with excess VPPs (5 uL) and rinsing to be sure we are recording signals indicative of membrane fusion to better compare the fusogenicities among different VOCs.

It is worth also pointing out that another way to maximize the signal is to increase the cell bleb composition (increase the number of receptors) in the hybrid SLB to capture more viruses and increase the change of electrical signal caused by fusion, however it is challenging to form an intact hybrid SLB with a higher concentration of native blebs. Increased bleb concentration leads to more imperfections in the bilayer in our current process and we have shown there is a limit even on glass surfaces to the amount of blebbed material that can be incorporated, when too many blebs obscure the available area for the fusogenic “rupture” vesicles to have enough room to interact with the surface.

Figure IV. A dose response of SLB impedance change to different amount of VPPs. Particles of the same concentration were added at the various volumes and the impedance measured. After the last addition of 10 μ L, the device was rinsed to remove excess particles in the medium.

5) It is not clear to me how one can unambiguously distinguish binding from fusion without help of microscopy images. The authors should provide more clear guidelines concerning this. I understand that the ΔR values are different. But are such values concentration independent? For example: Could many particles binding be interpreted as few particles fusing or vice-versa?

This is a great question. We have a long history looking at single particle binding and fusion events on SLB supported on glass surfaces using microscopy. We leverage this knowledge and approach to verify the electrical readout here. Indeed, optical signals leave little doubt when a binding event occurs and when a fusion event occurs as these have distinct fluorescent signals. Because the PEDOT:PSS is transparent, we verified this optically on the PEDOT:PSS coated glass slides too. For the electrical readout, it is more difficult to distinguish them; impedance reports an overall ensemble change in the electrical properties of the SLB during particle interaction. We see in Figure IV impedance can increase just by adding more particles, so the need to rinse is important to ensuring that the signal that is observed is due to specific interactions between the particles and the SLB. However, there is a way to ensure we are reading out changes due to fusion separately from binding as can be explained by considering the reviewer's question: how can we be sure the signals we obtain are either for binding alone of many particles, or fusion of few? It comes down to the execution of the experiment. For this virus, the dividing line between binding and fusion is the introduction of protease triggers and conditions that support fusion. For the late pathway, this is straightforward as the fusion inducing protease is added as a second step and works only when the pH is at the activating value for cathepsin. In this case, when particles are introduced and if they bind, impedance may go up. To induce fusion, the fusion triggers are added next. If fusion occurs, impedance will go up again. If fusion does not occur, impedance will stay the same as the binding signal after rinsing. Therefore, by knowing if the fusion trigger was added or not, and monitoring the subsequent

readout, one can deduce the signal arising from fusion separately from that obtained initially for binding. So if I have a lot of particles binding and no fusion, I will see an increase in impedance after the first step and no change after the trigger step. If I have few particles binding initially, I may see only a small impedance change, but if those particles fuse, I will see the change occur after I introduce the fusion trigger step.

For the early pathway, this is more difficult to distinguish. In this case, the protease is embedded in the bilayer and so available to cleave and trigger fusion immediately upon binding. In the way we have executed these experiments, it is not possible to decouple the two contributions to the signal. However, prompted by this reviewer's comment, we note here that there may be a way to do so in the future, mimicking the process that virologist use in cell-based assays to decouple binding from fusion. In the biological assay, cells are held at 4 degrees during the binding phase. Once binding is complete, cells are rinsed to remove excess particles, then the cells are warmed to 37 degrees to allow membrane fusion. One could imagine adding these temperature steps into our assay as well. A second approach that could also work in our assay would be to use an inhibitor of the protease that could be removed later, thus allowing the protease to cleave the spikes after the particles have bound. We did not attempt either of these assays, but it could be done if the experimentalist wanted to isolate fusion only. It is something worthy of pursuit in future work and we appreciate the reviewer's prompt.

For the work presented here, the important point to control when comparing different VOCs is to be as consistent as possible with the receptor and particle concentrations so that only changes in fusogenicity are captured by the impedance change. We have shown this is true both with technical replicates and biological ones, so we are confident that what we report here is repeatable and true, and that the differences we see across the VOC are real. This is further corroborated when comparing the independently reported biological traits of these VOCs and the alignment with our electrical results.

6) lines 176-177, "The particles count, provided by NTA analysis, allowed us to control the relative concentrations of the VPPWH1 and blebs used to assemble the SLBs." I believe there is a typo, "...VPPWH1 and blebs..." should perhaps say "...blebs and POPC vesicles..."? The VPPWH1 particles are not use to assemble SLBs... Also, the authors should provide more details on how they produce SLBs reproducibly with the same amount (%) of bleb material from the different bleb sources. The above sentence is all that is said on the subject and this control over SLB composition seems key to experiment reproducibility.

We thank the reviewer for catching that typo. We have corrected the sentences the reviewer referenced above in the main manuscript. To address the reviewer's concern about reproducibility and controlling bleb material amount, we endeavor to reproduce our bleb production the same manner each time we produce them, we characterize bleb size and concentrations after each batch is produced, and we adjust concentration when necessary to match conditions as closely as possible across runs. To give the reader a sense of this reproducibility, we have included additional western blots in Supplementary Fig. 1 with three biological replicates for both Vero E6 and TMPRSS2 blebs. These data indicate that across multiple biological samples the relative quantity of the receptors and proteases remains constant.

7) lines 144-145, "The fusogenic vesicles used in this work are reconstituted from purified 1-oleoyl-2-palmitoyl-sn-glycero-3-phosphocholine (POPC) lipids, which is the principal lipid component of mammalian and viral membranes." POPC lipids from Avanti are synthetic, not a purified natural product. While Avanti does mention that POPC is the most abundant lipid in their Egg PC extract, I believe the truth is that "oleoyl" and "palmitoyl" are the most abundant of the fatty acid chains and PC is the most abundant headgroup in most lipidomic studies I have seen of mammalian membranes, but never have I seen the intact POPC reported. Also, viral envelopes are derived from mammalian membranes and thus the lipid abundances are fairly similar. In short, this statement is a bit too "matter-of-fact" when it is more of a simplified extrapolation of observations. If I am wrong, I would love to see the citation(s) that supports the statement "POPC is the principal lipid component of mammalian and viral membranes".

Thank you. We have adjusted the wording in the referenced sentence.

Reviewer #3:

The manuscript entitled “Recreating the Biological Steps of Viral Infection on a Bioelectronic Platform to Profile Viral Variants of Concern” co-authored by Chao et al. describes a novel bioelectronics approach to study virus binding to the host cell membrane and subsequent fusion or endocytosis without using cells but membrane preparations instead. The authors have used a very timely example: SARS-CoV2 and variants. The authors reproduce for the known variants their individual fusogenic capacity and the mode of cell entry. The manuscript is thoroughly and clearly written, the scientific arguments are sound and the conclusions are strictly based on the experimental data. The manuscript will for sure find a lot of interest among the readership. This “infection on chip” approach provides a new and very practical new technology to perform very detailed studies on new virus species, mutants and mechanisms of infection. The reviewer is particularly impressed that the technology would not require genetically engineered cells DURING the assay – very different from most cell-based assays. Work with GMOs is only required to prepare the sensors but not for the final assays. This paved the way for a widespread use. Using membrane resistance as an indicator of viral fusion and assuming that fusion leads to an increase in resistance by a yet unknown mechanism, the data is without contradiction. Solely a clear explanation why R_m increases upon fusion and other questions regarding the electrical parameters of the membrane (see below) require more explanation and revision. I strongly recommend this manuscript for publication after my major and minor concerns have been addressed.

Major concerns:

Line 224: The measured spectrum of the SLB-free PEDOT electrodes differs from the simplifying illustration in the center of fig 2C. The experimental spectrum shows a minor dispersion between 1 kHz and 10 kHz so that it is not just a resistor-capacitor in series structure as the authors claim in line 225. Please phrase more cautiously in this respect. What is the origin of the central dispersion? It must have to do with the PEDOT/PSS on the electrode. The reviewer is aware that this is not in the center of this manuscript and certainly does not interfere with the story. But this dispersion is not due to any biological component and it is just masked upon coating the PEDOT-electrode by an SLB. It is still there and contributes to the readings. So it may even change upon acidification. What is the impact of the pH drop on PEDOT/PSS conductivity? Have the authors checked for the response of a pure PEDOT/PSS coated electrode upon pH shift? The reviewer understands that the PEDOT is covered and shielded by the SLB but is this truly complete?

We thank the reviewer for this comment. We believe the bump between 1 kHz and 10 kHz in the baseline signal comes from the non-ohmic contact between PEDOT.PSS and Au contact pad. This is because while pristine PEDOT.PSS film is widely considered as a highly conductive polymer, its conductivity is less than 10 S/cm^2 , which is about 45,000 times lower than that of Au. Hence, additives such as ethylene glycol are usually added to make the PEDOT.PSS film much more electronically conductive³ for electrode and transistor-based applications. The commercial PEDOT.PSS (PH 1000) we used in this study contains only 1.0- 1.3 wt. % solid content in water dispersion, and the recommended 5 vol% ethylene glycol can increase the PEDOT domain size and improve the conductivity of PEDOT.PSS film by 2 orders of magnitude.³ However, we anticipate that the larger PEDOT domains result in more heterogeneous film surface morphology, which is not ideal for SLB self-assembling. Hence, as described in Methods, no ethylene glycol was mixed with PEDOT.PSS suspension prior to PEDOT.PSS spin-coating, only (3-Glycidyloxypropyl)trimethoxysilane (GOPS) was added to crosslink the film.

To support our hypothesis that this bump is due to this effect, we fabricated two batches of devices with the same electrode dimension (0.0015 cm^2), one without ethylene glycol additive to the PEDOT.PSS (the same as devices we used in the manuscript) and one with 5 vol% ethylene glycol mixed with PEDOT.PSS suspension prior to spin-coating. Their EIS baseline signals are plotted in Figure V.

Figure V. A baseline comparison between PEDOT.PSS electrode with (red) and without (black) ethylene glycol.

As we can see, with the addition of ethylene glycol in the PEDOT.PSS film, the bump at the middle frequency range nearly disappeared, suggesting an improved electrical contact between Au and PEDOT.PSS, which is likely due to the enhanced PEDOT.PSS film electronic conductivity that results from the EG addition. Other than the bump at the middle frequency range, we do not observe significant impedance differences at the high frequency range, and we think this is because the thin PEDOT.PSS film (~ 80 nm) does not contribute much to the system resistance, which is dominated by the same buffer resistance in both cases. At the lower frequency range, the difference between the two is mainly coming from the variations in effective volume of the PEDOT.PSS films, possibly due to the addition of ethylene glycol changing the film thickness. We have added more detail in the **Methods** to clarify that no ethylene glycol was added hence the baseline shape is not an ideal “L” shape and we included Figure V in Supplementary information for reference.

We also would like to address the reviewer’s comments on the impact of pH on the conductivity of PEDOT.PSS. While we are aware that acid treatment can improve the PEDOT.PSS film conductivity^{4,5} as the reviewer correctly suggested, the acidity of PBS buffer we used ($\text{pH} = 5.5$) to activate cathepsin L does not alter the electrode baseline within the time course of our experiments (~ 30 mins) compared to its value at $\text{pH} = 7.4$. Below, we compare these conditions of the PEDOT.PSS electrode baseline by lowering pH from 7.4 to 5.5, as shown in Figure VI.

Figure VI. A comparison of PEDOT.PSS electrode baseline at different buffer pH.

After measuring a plain electrode baseline in PBS (pH = 7.4) plotted in black, the buffer pH was adjusted to pH = 5.5 using 1 M HCl and incubated for 30 mins before a new baseline was collected and plotted in red. As we can see, no significant signal changes are observed on the PEDOT.PSS electrode by lowering buffer pH in this range. We thus conclude the increase in SLB resistance at a lower buffer pH is due to SLB responding to a more acidic environment, and not PEDOT.PSS property change.

The authors present an equivalent circuit that contains the membrane resistance R_m and the capacitance C_m but the values for C_m are not reported at all. Why? Values of C_m are very potent indicators for the quality of the membrane preparation so it must be reported.

We thank the reviewer's comment. The reviewer is correct and we have emphasized this aspect now. Indeed, the SLB capacitance does not vary greatly, and the values do support the presence of a planar membrane on the surface, so we have revised our text and tables to include this now (Supplemental Table S1), together with electrode dimension information.

Figure 2C: According to figure 2C, the resistance of the SLB is $12.3 \text{ Ohm}\cdot\text{cm}^2$. And it is in the same order of magnitude in the other examples. This is very low and about two orders of magnitude lower than expected for lipid bilayers without any ion channels. Why is the specific membrane resistance so low? Is the SLB covering the electrode completely? Since (i) there is considerable scattering in R_m of different SLB preparations and (ii) R_m values are too low, it seems plausible that coverage is not complete, that there are gaps affecting the R_m readings. Any indication? The authors have to comment on the completeness of coverage. It is hard to judge in particular as the authors do not provide the electrode surface area – unless it escaped my attention.

We agree that we should have reported the electrode surface area, and it has now been included in Supplemental Table S1. To clarify the reviewer's points, when comparing a bilayer made of pure synthetic POPC lipids on the

PEDOT.PSS electrode to the resistance of hybrid SLB derived from native cell blebs and POPC vesicles, we always observe a relative lower resistance, which is consistent with our previously published work. We believe this is reasonable, as the native bilayer not only has ion channels that allow ions to pass, but the cell blebs have more complexity in composition resulting in lower packing density of the resulting bilayer and these, in turn, will lead to a lower electrical resistance. In addition, the cell blebs also have less propensity to rupture spontaneously, so it is not unexpected that there will also be more unruptured vesicles on the surface compared to the POPC vesicles alone, resulting in more defects and imperfections compared to the pure lipid SLBs, which will also be a contributing factor to lower resistance compared to the POPC case. We also agree that the SLB is clearly not defect free and we do not know how it may or may not “seal” the edge of the device. Presumably it is not sealing well at all, given that we are far away from an ideal black lipid membrane giga-ohm seal. However, perhaps remarkably, we have enough signal and specificity to observe the phenomena presented here, as well as in other investigations we have conducted, for example, on ion channel activity. Clearly PEDOT.PSS is a key factor here, though we do not understand completely why. PEDOT.PSS does lower the impedance of the electrode and that certainly helps boost the ability to take these measurements, but it is likely that there is something additional going on with the conducting polymer sensor itself that enables our ability to measure these changes in bilayer electrical properties.

We acknowledge the wide range of SLB resistance values we report in the manuscript, but we do not believe this is an issue, so long as the resistance values are of sufficient value (see response to reviewer #1 above on a similar point). To get a better handle on this ourselves, our lab has made significant progress in tuning SLB formation conditions on PEDOT.PSS. We share here a study we undertook to understand the distribution of resistances obtained from cell bleb SLBs created under the same experimental conditions. Figure I (above) shows a histogram of SLB made from HEK blebs with resistance values distribution across multiple devices and various electrode dimensions (a total of 56 experiments in total). In it, we see a normal distribution with the peak of SLB resistance value at around 10-20 $\Omega \cdot \text{cm}^2$.

This range of resistance values is indeed lower than black lipid membranes and even tethered membranes, but we note here that the formation of our bilayers proceeds by vesicle fusion, driven mainly by van der Waals interactions between the plasma-treated PEDOT.PSS and the lipid headgroups. The advantage of this assembly approach is it is free of organic solvent, which permits the incorporation of native cell membrane components in the SLB. This tether-free assembly also promotes mobile constituents (confirmed by our FRAP results in Figure S4-6, and Figure S10) and can thus support membrane fusion between SLB and viral envelopes highlighted in this work. Finally, we note that the presence of PEDOT.PSS coating on the electrode certainly complicates the vesicle fusion self-assembly process compared to a pristine glass surface, but it offers many excellent features that make it worth the effort to optimize the assembly process including reduction in gold electrode impedance to vastly improve signal-to-noise and allow biorecognition events, inertness and biocompatibility, a cushioning effect that fosters bilayer constituent mobility, and optical transparency that enables additional independent characterization via microscopy.

Line 246: The increase in resistance upon viral fusion is unexpected and counter-intuitive. The membrane surface area increases upon fusion of SLB and virus membrane so that the area-specific resistance should decrease. The authors speculate that additional biomacromolecules from the viral membrane get transferred into the SLB or gaps are filled (see above). To the reviewer's understanding, such a transfer of biomolecules should lead to an increased membrane capacitance C_m . A protein-free bilayer has about $0.7 \mu\text{F}/\text{cm}^2$ whereas mammalian cell membranes with regular protein content show app. $1 \mu\text{F}/\text{cm}^2$ due to the presence of protein with higher dielectric constant. It would support the discussion and help the conclusion to report the C_m values next to R_m values. Also: A rather simple experiment may lead to more insight and I am almost sure that the authors have done it: titration experiments with sequentially increasing virus load on one and the same SLB. The more VPPs fuse, the more the electrical signals should change.

We thank the reviewer's prompt to elaborate on this point. We now report the capacitance of SLBs in Supplemental Table S1. The cell blebs derived SLBs have a calculated capacitance ranging from 0.84 – 1.86 $\mu\text{F}/\text{cm}^2$, with most of them at around 1 $\mu\text{F}/\text{cm}^2$ as the reviewer suggested. This value falls in the expected range of a planar bilayer of thickness 4-5 nm. However, it is an ensemble average of the entire surface, and we acknowledge that it would also capture any contributions that result from defects like unruptured vesicles and incomplete bilayer formation (such as holes). As such, it is difficult to interpret exactly what variations are telling us.

Regarding the point that the bilayer surface area should increase after pseudoparticle fusion, we simply don't know. On one hand, that seems obvious, however, the device is confined to a well (with a step height of ~120 nm) and it is possible that the lipid bilayer forms across the entire surface of that well, including the parts that are insulated, resulting in little availability for spreading out in a greater sense. What we believe is actually more likely the case is that there is likely local pinning of the bilayer due to irregularities in the PEDOT.PSS film and that when a virus fuses, the lipids in that localized area pack together more tightly to accommodate that influx of material. We rationalize this conclusion because we don't observe a large change in membrane capacitance, rather, we observed a significant increase in membrane resistance.

We did not carry out the titration experiment in exactly the way that the reviewer suggested, rather we ran it to ensure we were maximizing our signal by working in excess pseudoparticles (See Figure IV above and response to the reviewer there). Our goal was to ensure we captured as many particles as possible on our surface by occupying maximum receptor sites and thereby ensure we obtain the most fusion signal possible and the best signal to noise and most fairly compare fusogenicity trends across VOCs.

Why are the electrical resistances of SLBs with identical composition so different from experiment to experiment? For example: Vero SLB + ACE2 + TMRSS2 is 18.7 $\text{Ohm}\cdot\text{cm}^2$ or 11.3 $\text{Ohm}\cdot\text{cm}^2$. Is the SLB preparation not yielding complete membrane coverage of the electrode? Are the electrode edges completely covered?

We addressed the question about the distribution of SLB membrane resistance in our previous responses and in Figure I. Regarding the complete coverage of electrode edges by lipid bilayers raised by the reviewer, we simply don't know what is happening at the edge of the well. At this point in our research and work, we have optimized our device configuration and currently the step height at the electrode edge between insulating SiO_2 layer and PEDOT.PSS surface is about 120 nm (200 nm of SiO_2 minus the height of PEDOT.PSS layer, which is about 80 nm). We believe a reduced step height helps form a more complete SLB up to the electrode edge and helps to 'seal' the overall sensor patch as completely as possible, thus we believe the influence of the SLB edge is minimized to the extent possible. What we are sure of is that our devices yield reliable data and trends despite variation that we see in SLB resistance from batch to batch. As noted above, we have some tolerance in the sensor to sense the events over a range of initial resistances. While we strive to obtain the highest quality SLBs and resistance values possible, practically, we have to live with non-idealities that limit this. Nonetheless, we are fortunate that these issues do not preclude our ability to take reliable measurements, as we believe we have demonstrated here with virus fusion, and we have also shown in other work on ion channels. In our conclusion, the devices are sensitive enough to work well for our purpose, despite material defects in the SLBs.

Minor Concerns:

Line 177: Supposedly „concentrations of the VPPwh” should read “concentration of vesicles”.

We have fixed this error.

Line 223: Please reword “.. the circuital response to alternating voltage with changing frequency...” maybe like “frequency-dependent impedance....”.-

We have reworded the sentence as per the reviewer's suggestion.

Line 259: The data in figure 3b suggest that the resistance increase due to late entry is significantly bigger compared to the early entry. The data pooled in figure 3c indicate the opposite: late stage entry increases the resistance less than early stage entry. Isn't this contradictory?

The data shown in 3b and 3c have a similar % resistance change. Data in 3b show an increase from 13.1 to 19.9 and in 3c show an increase from 23.3 to 36.7 upon fusion. On average, the percent change in resistance was observed to be slightly higher for the early entry pathway, likely due to the localization of both the ACE2 receptor and TMPRSS2 protease in the membrane.

Line 354: Rephrase „surface electrostatics measurement“ to maybe “dielectric measurements”.

We have changed the wording of the sentence.

References

- 1 Shuai, H. *et al.* Attenuated replication and pathogenicity of SARS-CoV-2 B.1.1.529 Omicron. *Nature* **603**, 693-699, doi:10.1038/s41586-022-04442-5 (2022).
- 2 Kim, Y. H. *et al.* Highly Conductive PEDOT:PSS Electrode with Optimized Solvent and Thermal Post-Treatment for ITO-Free Organic Solar Cells. *Advanced Functional Materials* **21**, 1076-1081, doi:<https://doi.org/10.1002/adfm.201002290> (2011).
- 3 Rivnay, J. *et al.* Structural control of mixed ionic and electronic transport in conducting polymers. *Nature Communications* **7**, 11287, doi:10.1038/ncomms11287 (2016).
- 4 Kim, N. *et al.* Highly Conductive PEDOT:PSS Nanofibrils Induced by Solution-Processed Crystallization. *Advanced Materials* **26**, 2268-2272, doi:<https://doi.org/10.1002/adma.201304611> (2014).
- 5 Zhang, L. *et al.* The Role of Mineral Acid Doping of PEDOT:PSS and Its Application in Organic Photovoltaics. *Advanced Electronic Materials* **6**, 1900648, doi:<https://doi.org/10.1002/aelm.201900648> (2020).

REVIEWERS' COMMENTS

Reviewer #1 (Remarks to the Author):

This reviewer has no other question.

Reviewer #2 (Remarks to the Author):

I am satisfied with the authors response to all reviewer's comments and fully endorse the publication of this manuscript in its current form in Nature Communications.

Reviewer #4 (Remarks to the Author):

As mentioned by the Rev. 3 themselves, these concerns are not crucial. The authors have addressed well the polymer's characterization requested by the reviewer. Their data is convincing and the trend is reasonable. The model they use is often used in the community and, despite the lack of a general agreement, it is a good approximation. Precise estimation of electrochemical values is a hard task but overall the model shows a decent trend as shown in the fit.

All the technical explanations help the discussion and respond exhaustively to each point.